# Knapping tools in Magdalenian contexts: New evidence from Gough's Cave (Somerset, UK)

**Silvia M. Bello** [1]*, **Lucile Crété**[1], **Julia Galway-Witham**[2,3], **Simon A. Parfitt**[1,4]*

**1** Centre for Human Evolution Research, Department of Earth Sciences, Natural History Museum, London, United Kingdom, **2** Department of Anthropology, New York University, New York, NY, United States of America, **3** New York Consortium in Evolutionary Primatology, New York, NY, United States of America, **4** Institute of Archaeology, University College London, London, United Kingdom

055 These authors contributed equally to this work.

\* s.bello@nhm.ac.uk (SMB); s.parfitt@nhm.ac.uk (SAP)

**Data Availability Statement:** All relevant data are within the paper.

**Funding:** S.M.B, L.C., J. G-W., S.A.P work was funded by the Calleva Foundation; J. G-W.

## Abstract

Our knowledge of the recolonization of north-west Europe at the end of the Last Glacial Maximum depends to a large extent on finds from Gough's Cave (Somerset, UK). Ultra-high resolution radiocarbon determinations suggest that the cave was occupied seasonally by Magdalenian hunters for perhaps no more than two or three human generations, centred on 12,600 BP (~14,950–14,750 cal BP). They left behind a rich and diverse assemblage of Magdalenian lithic and osseous artefacts, butchered animal bones, and cannibalised human remains. The faunal assemblage from Gough's Cave is one of the most comprehensively studied from any Magdalenian site, yet new and unexpected discoveries continue to be made. Here, we record previously unrecognized flint-knapping tools that were identified during a survey of the Gough's Cave faunal collection at the Natural History Museum (London). We identified bones used as hammers and teeth manipulated as pressure-flakers to manufacture flint tools. Most of the pieces appear to be *ad hoc* (single-use?) tools, but a horse molar was almost certainly a curated object that was used over an extended period to work many stone tools. This paper explores how these knapping tools were used to support a more nuanced understanding of Magdalenian stone-tool manufacturing processes. Moreover, we provide a standard for identifying minimally-used knapping tools that will help to establish whether retouchers and other organic stone-working tools are as rare in the Magdalenian archaeological record as current studies suggest.

## Introduction

Over the past 25 years, studies of Palaeolithic bone collections have increasingly recognized that bones, teeth and antlers were used routinely as knapping tools to manufacture stone tools (Fig 7 in [1], p.114). Understanding how these tools were used complements the more traditional approaches for interpreting early stone tool technologies, and they provide a new perspective for discussions about the technical abilities of early human populations [e.g. 2].

Currently the earliest examples of organic hammers are from Bed II at Olduvai Gorge, Tanzania [3]. Here, a giraffe astragalus and an elephant patella appear to have been used as

contribution was also funded the National Science Foundation Graduate Research Fellowship. The funders had no role in study design, data collection and analysis, decision to publish, or preparation of the manuscript.

**Competing interests:** The authors have declared that no competing interests exist.

knapping percussors, possibly in the manufacture of handaxes or cleavers. By the Middle Pleistocene in Europe, a wide range of bone types have been linked to flint-knapping tasks. For example, the organic knapping tools from Boxgrove (UK, 500,000 BP) include distal humeri of deer and bison that were used as knapping hammers, and expertly worked antlers of red deer and giant deer that are identical to soft hammers used by modern-day flint knappers [4–8].

Middle Palaeolithic sites have produced substantial quantities of bone fragments used as *ad hoc* knapping tools ('retouchers'). These are exemplified by examples from Neanderthal occupation horizons in French cave sites, such as La Quina, where these tools were first recognised by French archaeologists in the early years of the twentieth century [9, 10]. Later stone working innovations during the South African Middle Stone Age (~75,000 BP) include heat-treatment of silcrete and pressure flaking in the manufacture of Still Bay bifacial points [11, 12]. Upper Palaeolithic knappers utilized similar techniques to deliver the high degree of control required to detach blades and bladelets from cores and to work these blanks into a wide variety of often finely-worked and standardized implement types, such as projectile points, awls, borers, burins, notches and end scrapers. Many of these lithic tools were used to process organic raw materials. Although wooden artefacts are extremely rare from the Upper Palaeolithic in Europe, carefully-fashioned objects made of ivory, bone and antler are one of the 'hallmarks' of Upper Palaeolithic industries in Europe, particularly during the Magdalenian, when osseous technologies were probably at their most developed.

The Magdalenian was a widespread Late Upper Palaeolithic cultural tradition that can be found over a large part of Late Glacial Europe, spanning the period following the Glacial Maximum (~ 18,000 BP) until it was supplanted by other traditions between about 13,000–11,000 BP. One of the characteristics of the Magdalenian is the extensive use of ivory, bone and antler that was carved, incised and whittled to fashion symbolic artefacts such as beads, figurines and abstract engravings, as well as utilitarian objects such as eyed needles, beveled bone points, barbed harpoons, spear throwers, wedges, awls and polishers. Studies of Magdalenian artefacts have mainly concentrated on stone tools and osseous artefacts and, as a consequence, few workers have considered in any detail the role of knapping tools in Magdalenian stone-working. Studies of Magdalenian stone tools show that blades were detached with a soft hammer. For example, Jacobi's [13] analysis of the flint assemblage from Gough's Cave (Somerset, UK) has shown that a soft hammer mode of blade production accounts for nearly all of the knapped pieces. With this in mind, it is perplexing that organic (soft) knapping tools appear to be rare in Magdalenian contexts.

Here we present a hitherto unpublished group of Magdalenian organic knapping tools from Gough's Cave. Our analysis and interpretation of these finds has three main aims. First, we provide descriptions of the pieces with a focus on diagnostic features that allow their recognition as knapping tools. We apply a range of imaging and analytical techniques to provide an in-depth analysis to deduce the 'biography' of the knapping tools, namely to identify how the raw materials were obtained and modified for use, how the tools were used in the knapping process, and to determine how they were modified after they were discarded or lost. Second, we examine the spatial distribution of the knapping tools and flintwork to identify activity areas within the cave and describe the lithic *chaîne opératoire* by assessing how the different types of knapping tools were used. Finally, we compare the Gough's Cave knapping tools with those from other well-studied Magdalenian sites, to throw light on an aspect of Magdalenian bone technology that is rarely considered in the archaeological literature.

## Archaeological context and age

Gough's Cave is the richest Upper Palaeolithic site in the British Isles in terms of human remains, flint, organic artefacts and butchered faunal remains. The cave opens on the southern

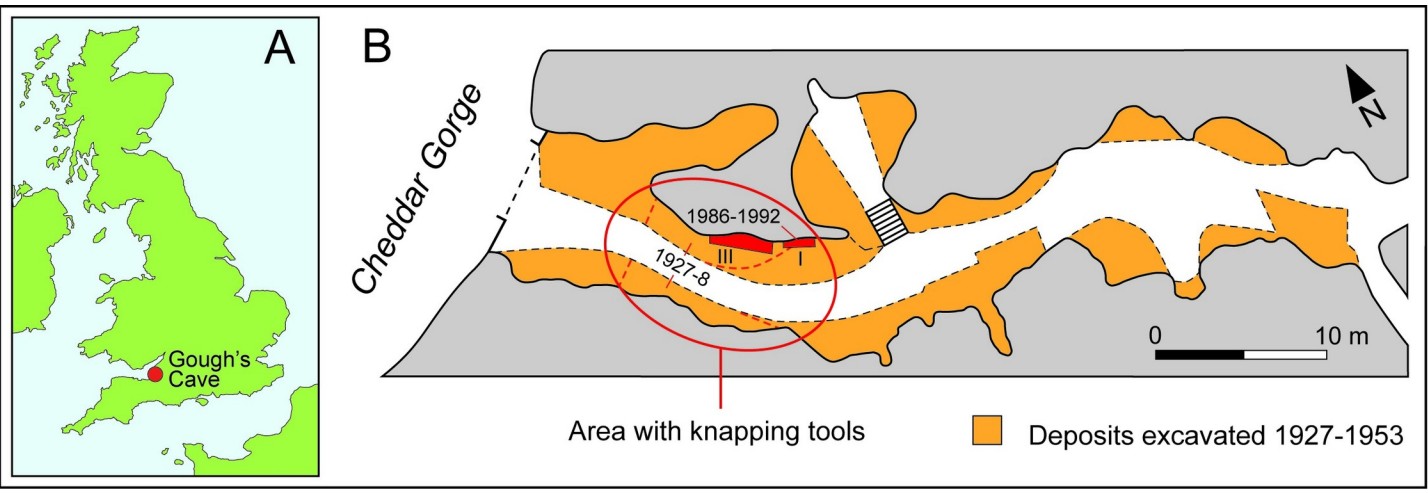

**Fig 1.** A. Location of Gough's Cave. B. Plan of the outer part of Gough's Cave showing where the knapping tools were recovered. Five of the eight knapping tools from Parry's excavation were found in November 1927, and the other specimens recorded for that year, as well as the phalanx from the 1987 excavation (Area I), were found in the same area of the 'entrance' (bounded by the red oval). Plan based on [13–15].

side of the entry to Cheddar Gorge, Cheddar, in Somerset (51˚ 15'N, 2˚ 45' W. National Grid Reference: ST 46705391.—c. 30 m above sea level; Fig 1). At the time that the cave was occupied by Magdalenian hunters, the gorge was situated at an ecological and topographic divide between lowland marshes, lakes, and floodplains of the Somerset Levels and the Bristol Channel, and a high plateau of the Mendip uplands, which peak at 260 m above modern sea level. Located at the mouth of this narrow, steep-sided limestone gorge, Gough's Cave was ideally sited with access to a diversity of habitats. More importantly, perhaps, the gorge provided an ideal conduit for driving and trapping horses moving seasonally between these two zones.

Gough's Cave has a long history of investigations that stretches back to the discovery and opening of the cave as a tourist attraction by Richard Cox Gough in the 1890s [16]. The remaining deposits at the front of the cave were almost completely excavated by R.F. Parry over several winter seasons between 1927 and 1932 [17–21]. A.V. Painter continued work up to 1953, which saw the removal of the distal part of the wedge of Late Glacial sediments. The most recent investigations were directed by the late Roger Jacobi (University of Lancaster, University of Nottingham) and Andy Currant and Chris Stringer of the Natural History Museum (London), who undertook exploratory work in 1986–7. This excavation identified exceptionally rich remnants of Magdalenian deposits surviving in the narrow gap between the concrete floor and the cave wall [22]. Further excavations along the north wall of the cave were undertaken in 1989–90 with two productive trenches (Area I and III) that investigated a continuation of this deposit [15].

The collection of lithics from Gough's Cave is impressive for a British Late Upper Palaeolithic site with numerous flint tool-types based on blades and bladelets [13, 23–27]. Bone, antler and ivory also served as raw materials for artefact manufacture. The organic tools and other objects include an intriguing mix of domestic and ritual objects, such as perforated *bâtons* [28], mammoth ivory javelin heads (bevel-based point or 'sagaie'), 'blanks' of swan and hare bone from which needles were cut, awls, a needle, fox tooth beads, sea shells, incised ivory, amber and scratched pebbles [29].

The archaeological finds indicate that the site functioned as a short-lived, multi-activity seasonal camp (occupied in a series of intermittent visits in the summer and winter) with a focus on horse and red deer hunting [13]. This interpretation is supported by the human age

structure that included infants and an older child, as well as older adults, implying that the site was occupied by family groups [30, 31]. A particularly intriguing aspect of the human bone assemblage is the evidence of butchery of the cadavers with traces from scalping, intensive cleaning of the bones to remove edible soft tissue, fracturing of bone shafts to extract marrow, and chewing and consumption of spongy bone. Heads were also modified to make skull cups, and a radius is marked with a chevron motif [31–36]. These traces strongly suggest that cannibalism was ritual in nature, although other scenarios, such as starvation cannibalism, have been considered [e.g. 37, 38].

Bayesian modeling of the latest ultrafiltered AMS radiocarbon determinations on the human remains, butchered animal bones and artefacts show that the Magdalenian occupation of Gough's Cave began at the same time as the rapid climatic amelioration that marks the transition from Greenland Stadial 2 (GS-2) and the start of Greenland Interstadial 1 (Gl-1e), equated with the Bølling Chronozone in the European terrestrial record [39, 40]. The dates display a remarkably tightly clustered group with a mean value of 12,600 BP (~14,950–14,750 cal BP) indicating that the Magdalenian occupation lasted for as little as two or three human generations [40].

Summer temperatures during the Magdalenian occupation phase may have been close to those of today, although winters were substantially colder. Pollen, mammalian and avian evidence indicates a landscape dominated by steppe-like vegetation and a patchy growth of birch, hazel and alder in sheltered areas of the gorge [41–44].

Pertinent to this paper are the recent taphonomic studies of the faunal remains [28, 31–36]. This work has been at the forefront of applying new imaging methods to the study of taphonomic and archaeological problems [32]. We now routinely use environmental scanning electron microscopy (eSEM), 3D optical imaging [36, 45], energy dispersive x-ray analysis (EDX) and element mapping (this paper), and computed tomography [28] to study the faunal remains. These methods have contributed significantly to the taphonomic interpretation of the prey animals and the human remains, and the use of these techniques was critical in recognizing and interpreting the bone knapping tools described in this paper. The discovery of this previously unrecognized category of bone tool in such an intensively studied collection [34, 46] casts doubt on any suggestion that the published taphonomic investigations have exploited the full potential of the Gough's Cave collection.

## Material and methods

### Identifying the knapping tools and methods of analysis

The state of preservation of the faunal remains from Gough's Cave stored at the Natural History Museum, London (NHM), was determined according to bone fragmentation and soundness of cortical surface using criteria defined by Behrensmeyer [47] and Bello et al. [48]. Human induced modifications were classified as slicing cut-marks [49], scrape-marks, chop marks (*sensu* [50]), and percussion damage [50–53]. Diagnosis of chewing marks was based on morphological features and their location on bones [54–56].

Knapping marks are caused when a softer organic material comes into contact with a hard stone during knapping. The characteristics of these marks have been comprehensively studied and considered to be clearly distinguishable from other damage. For our research, the identification of organic retouchers was based on the following diagnostic characteristics:

1. Presence of gouges, pits and scores. These indentations are characteristically angular in plan with internal microstriations [2, 9, 57–64]. Typically, these marks are concentrated in clusters where the percussor was struck repeatedly against the edge of the stone tool during knapping [65, 66].

2. Presence of lithic fragments embedded within the knapping marks. These inclusions, although not regularly documented, have been observed in retouchers and percussors used during experimental knapping [67, 68], as well as in some archaeological examples [e.g. 1, 2, 63, 69, 70].

3. Presence of tool-edge scratches, or 'sliding striations' [59]. These marks are small abrasions that are formed when the edge of a lithic tool slides across the surface of the knapping tool during percussion or pressure flaking [59, 64].

The knapping tools from Gough's Cave were identified during a review of the large mammal remains (approximately 300 human and non-human specimens). Promising examples were identified by eye and an initial examination under a hand lens and binocular microscope was undertaken to confirm the identifications. The knapping tools identified in this survey include two teeth, four distal metapodials, a complete metatarsal and a proximal phalanx of horse and a distal metatarsal of red deer (Table 1).

The breakdown of specimens examined and the proportion used as knapping tools is as follows:

1. Horse incisors–n = 29 (excluding teeth in mandibles/maxillae), 1 knapping tool (3.4%)

2. Horse upper molars–n = 24 (excluding teeth in mandibles/maxillae), 1 knapping tool (4.2%)

3. Horse distal metapodials–n = 14, 5 knapping tools (35.7%)

4. Horse first phalanges–n = 22, 1 knapping tool (4.5%)

5. Red deer distal metapodials–n = 7, 1 knapping tool (14.3%)

The knapping tools listed in Table 1 include 'text-book' examples as well as 'atypical' specimens. Their locations within the cave (Fig 1B) and the circumstances of their discovery are noted in Table 2.

The Gough's Cave knapping tools are described below and the surface features are illustrated using optical imaging and electron microscopic methods. Imaging of the surfaces was first undertaken using an Alicona Infinite Focus optical microscope. This is a focus variation microscope (FVM) that we used to generate three-dimensional digital models of the surface features. Details of these features were then observed and recorded using a scanning electron microscope (SEM, the JEOL-IT500) operated under variable pressure mode. This SEM is fitted with an energy-dispersive X-ray (EDX) spectroscope (Oxford Instrument X-Max 80 Silicon Drift Detector with INCA software), which we used to identify the lithic inclusions. Both

**Table 1. List of teeth and bones from Gough's Cave used as knapping tools.**

| Museum no. | Anatomical element | Species | Figure |
|---|---|---|---|
| NHM PV M50064 | Upper left third molar | *Equus ferus* | 3 |
| NHM PV M49811 | Upper left central incisor | *Equus ferus* | 4 |
| NHM PV M49934 | Right metacarpal, distal end | *Equus ferus* | 5 |
| NHM PV M50024 | Right metacarpal, distal end | *Equus ferus* | 6 |
| NHM PV M50000 | Left metacarpal, distal end | *Equus ferus* | 7 |
| NHM PV M49873 | Right metatarsal | *Equus ferus* | 8 |
| NHM PV M50025 | Left metatarsal, distal end | *Equus ferus* | 9 |
| NHM PV M49847 | Left metatarsal, distal end | *Cervus elaphus* | 10 |
| NHM PV UNREG 3482 | Proximal phalanx | *Equus ferus* | 11 |

**Table 2. Locations and circumstances of discovery, and radiocarbon dates for the knapping tools.**

| Museum no. | Anatomical element | Find details | Sampling for radiocarbon determinates [39, 40] and aDNA |
|---|---|---|---|
| NHM PV M50064 | M$^3$ | Parry excavation November 1927 (presented 1928), unstratified | |
| NHM PV M49811 | I$^1$ | Parry excavation 1927 (presented 1928), spit 9 | |
| NHM PV M49934 | Metacarpal | Parry excavation November 1927 (presented 1928), spit 14 | |
| NHM PV M50024 | Metacarpal | Parry excavation (presented 1928), Cave Earth/Breccia unit, spit 18 | OxA-464 (AC), 12,470 +/- 160 yr BP; resampled as OxA-17832 (AF), 12,415 +/- 50 yr BP (NHM dating sample P 0826). Drilled hole |
| NHM PV M50000 | Metatarsal | Parry excavation November 1927 (presented 1928) | |
| NHM PV M49873 | Metatarsal | Parry excavation November 1927 (presented 1928), spit 13 | aDNA sample (rectangular cut hole) |
| NHM PV M50025 | Metatarsal | Parry excavation (presented 1928), spit 18 | aDNA sample (rectangular cut hole filled with plaster) |
| NHM PV M49847 | Metatarsal | Parry excavation November 1927 (presented 1928), spit 13 | OxA 466, 12,800 +/- 170 yr BP; OxA-16378 (AF), 12,515 +/- 50 yr BP (NHM dating sample P 0859). Drilled hole and piece cut from broken end |
| NHM PV UNREG. 3482 | Phalanx | BM(NH) excavation 1987. 1242.0, field number 23 | |

Specimens have been radiocarbon dated using accelerator mass spectrometric (AMS) techniques (OxA). AF denotes ultrafiltered gelatine determinations; AC determinations are for decalcified bones with hydrolysis of the residue and treatment with activated charcoal before separation of the amino acids using cation-exchange columns [39, 40].

instruments are housed at the Natural History Museum, in London (UK). The combination of these imaging techniques is particularly suitable for the analysis of surface modifications because it offers 3D visual reconstructions of the object surface (Alicona), and it enables recognition of fine details only visible at high resolution (SEM) as well as of exogenous material embedded in the specimen (SEM with associated EDX system) [71, 72].

## Abbreviations

BM(NH)–British Museum (Natural History) now the Natural History Museum, London (NHM). All the material is in the collection of the Department of Earth Sciences, Natural History Museum, London. The full registration prefix 'NHM PV M' is abbreviated in the text as 'M'; the specimen prefixed 'UNREG.' is currently on loan from the Longleat Estate.

No permits were required for the described study, which complied with all relevant regulations.

## Results

Bone preservation of the Magdalenian bones at Gough's Cave is almost perfect and surface marks, such as cut marks, chew marks from carnivores and even superficial tooth marks made by humans [31, 34] can readily be identified. Although some of the bones have been marked by weathering, trampling, plant roots and animal chewing, this damage is minimal and the knapping features are clearly visible. Further cleaning of the bone surfaces was not undertaken and details of some features are obscured by adhering sediment. Some features are covered by museum labels and four of the retouchers have been drilled or cut to remove samples for radiocarbon determinations, relative dating or isotope analyses before they were recognized as bone tools.

Among the faunal remains from Gough's Cave, we identified two teeth and seven bones that exhibit atypical percussion damage (Table 1). The wear marks displayed by these specimens are of a standard type and consist of angular indentation (pits, scores and gouges) that are often associated with microstriations and flaking, chipping and cracking adjacent to the impact areas; internally the bases and sides of the percussion marks show faint striations running transverse to the groove. The percussion features are found as isolated or dispersed features, but more commonly they occur in discrete groups where the impact features overlap and inter-cut each other. The marks exhibit similar patterning of length, shape, and orientation. Moreover, rare examples were found with microscopic lithic fragments embedded within the percussion features. This combination of features is the hallmark of bones and teeth that have been used as knapping tools in stone tool manufacture.

## Radiocarbon dating of the knapping tools

Two of the knapping tools have AMS ultra-filtered radiocarbon determinations (Oxford Radiocarbon Accelerator Unit). These are included in the series of dates on humanly modified faunal remains and human remains that were used to determine the short duration of the Magdalenian occupation at the site [39, 40]. Details of the dating results on the knapping tools are provided in Table 2. These dates are important as they link the knapping tools with the oldest radiocarbon ages for Late Palaeolithic use of the cave, rather than the second episode of Palaeolithic occupation linked to the sparser occurrence of Federmessergruppen ('Azilian') artefacts.

## Teeth used as knapping tools

**Upper third molar, *Equus ferus* (M50064).** This is a complete horse molar in early wear recovered by Parry in November 1927. The specimen is recorded as 'unstratified', but its condition and dental size are entirely consistent with its origin in the Magdalenian horse assemblage [41, 73]. The surface of the tooth exhibits a plethora of microscopic knapping pits, scores and scratches which mark the enamel on four sides of the tooth (Fig 2A, 2B and 2J-2N). In particular, intense wear is concentrated on the flatter (mesial) surface of the molar, near the middle of the crown and towards its base. In these zones, repeated micro-chipping of the enamel has hollowed-out a dish-shaped depression. Microscopic examination shows that the enamel in the working areas is roughened by brittle fracturing and chipping of the enamel. Micro-chipping extends beyond the edges of depressions and becomes increasingly diffuse across the surrounding enamel surface (Fig 2F). On examination in the SEM, these depressions show a multitude of intersecting, obliquely-oriented gouges and chipped enamel. Minute chips of flint were found embedded in the enamel; some are easily discernible in the SEM images (Fig 2H), but the EDX analysis also picks out high concentrations of silica in other gouges that may be chips of flint which are more deeply embedded in the enamel or dentine (Fig 2I). Although fragments of lithic debris are not uncommon in bones used as knapping tools [2, 63, 67, 70], we believe this is the first record of lithic debris embedded in a tooth used to work flint.

The area of intense attrition on the medial face is associated with flaking that has removed the mesostyle along the lower two-thirds of the tooth. The flake scars are scallop-shaped and originate from pressure directed against the mesial face, which has forced flakes of enamel from the buccal side of the tooth (Fig 2C and 2D). Cracks in the enamel may be faults signalling incipient flaking (Fig 2E). Bello and Galway-Witham ([72], Fig 6, p.24) examined this tooth with computerised tomography. Using this method, they showed that the flaking extended into the underlying dentine. The enamel surrounding the clusters of knapping marks also exhibits transverse micro-striations (Fig 2G).

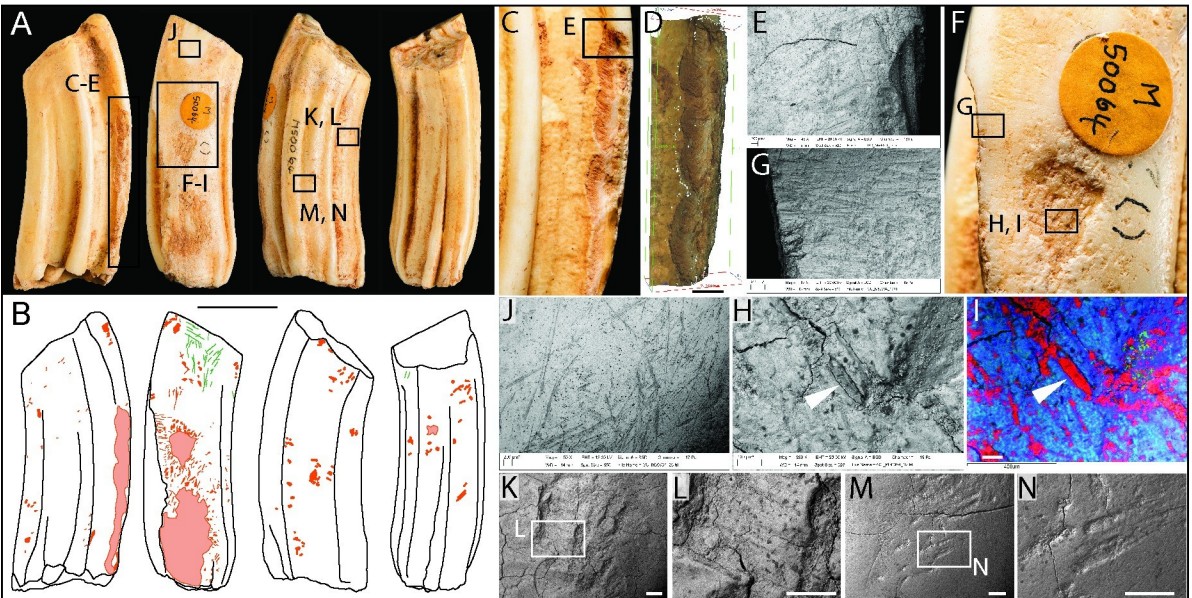

**Fig 2.** Photographs (**A**) and drawings (**B**) of horse molar M50064, showing knapping damage (pink) and scratches (green) in labial, mesial, lingual, and distal views (Scale = 50 mm). SEM micrographs, Alicona images and macro-photograph detailing: (C-D) chipped enamel along the mesostyle; (E) transverse parallel abrasions associated with flaked enamel; (F) macro-photograph of the working area on the mesial surface, showing minute punctiform features, transverse micro-abrasion and a large depression where the focus of pressure-flaking has abraded the enamel to form a bowl-shaped depression. This area has many pits and scores with two dominant alignments at right angles to each other; (G) detail of transverse micro-striations associated with the areas of concentrated knapping features on the mesial surface; (H) SEM image and (I) EDX element plot highlighting fragments of flint embedded in the enamel. Red = silicon (flint and sediment), blue = calcium (enamel); (J) example of random striations from contact with a stone tool. Although these marks occur over much of the surface of the tooth, they are more easily discernible on approximal wear facet; (K-L) gouge with flaked enamel and internal parallel striations; (M-N) indentations that appear to be compression features or shallow abrasions.

Occasional isolated or small clusters of indentations occurring on the lingual and distal faces are illustrated in Fig 2K–2M. A variety of marks were observed, including gouges with internal microstriations (Fig 2K and 2L) and shallower depressions that appear to have been impressed (Fig 2M and 2N). The latter marks have smoother cross-sections, suggesting they were created by pressure that has deformed or abraded the enamel.

Taking into account the microscopic characteristics and distribution of the marks, it is possible to decipher how this tool was used. Although the tooth could have been used to work flints by direct percussion, it is more likely that it was used as a pressure flaker. Semenov ([57], Fig 95–5) shows similar pressure flakers used by gripping the tool between the palm and fingers and levering it against the lithic tool-edge, in the manner of using a bottle-opener to lever a bottle top. Similar modifications have also been observed in recent experimental work on pressure flakers by Nami and Scheinsohn [74], Armand and Delagnes [75], d'Errico et al. [76], Mozota [77], and Doyon et al. [78]. It is also possible that some of the most superficial semi-parallel scratches were produced during preparation of the margins of the lithic tools before retouching it [cf. 66].

**Incisor,** *Equus ferus* **(M49811).** This horse incisor was found in spit 9 of the 1927 excavation. Knapping marks are concentrated on the buccal face of the crown (Fig 3A and 3B) and include transverse gouges and angular pits located near the base of the crown (Fig 3C–3F) and vertically oriented scratches that extend from the gouges towards the incisal surface (Fig 3G–3I). The superficial abrasions of the enamel with multiple striations are consistent with a lithic tool-edge sliding across the enamel surface. These abrasions could have been produced during

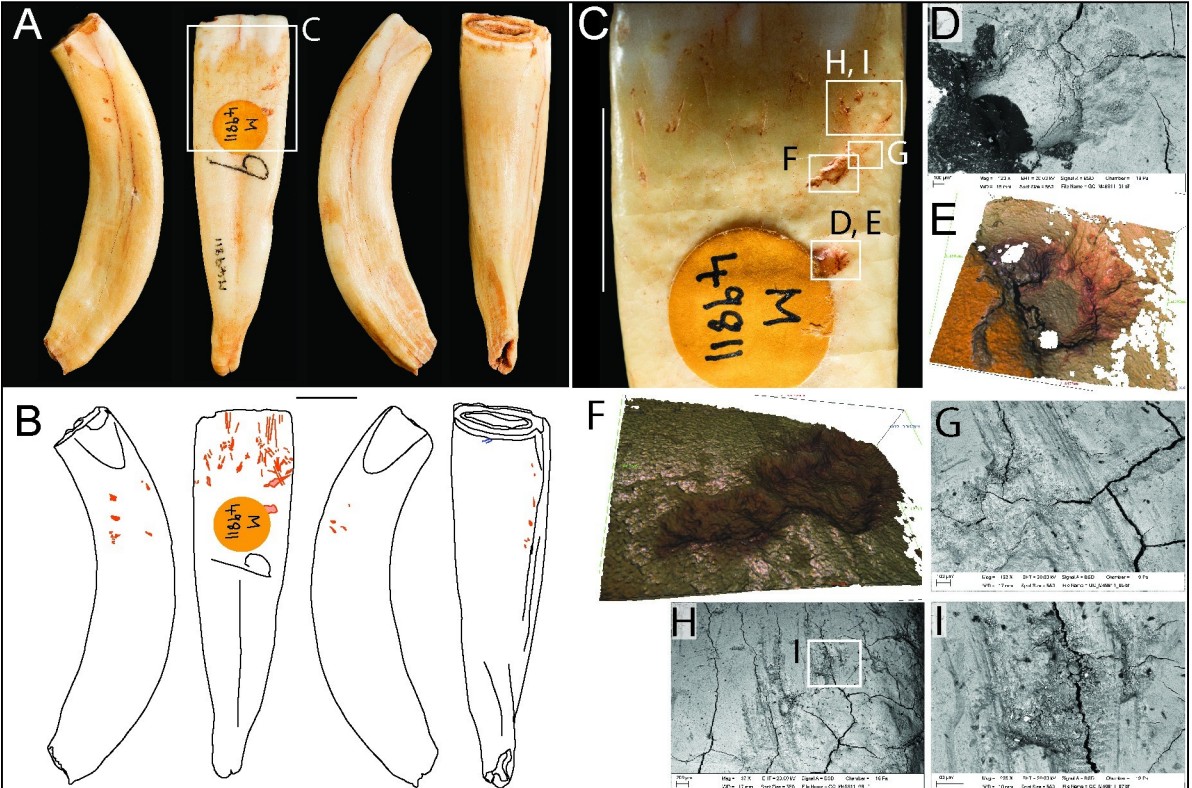

**Fig 3.** Photographs (**A**) and drawings (**B**) of horse incisor M49811, showing knapping damage (pink) in mesial, labial, distal, and lingual views (Scale = 10 mm). SEM micrographs, Alicona images and macro-photograph detailing: (**C**) close-up of knapping marks near the crown-root junction on the buccal surface; (**D-E**) isolated pit; (**F-G**) score with associated tool-edge scratches; (**H-I**) tool-edge scratches and micro-pits. Pits and scores are concentrated near the crown-root junction, whereas vertically-aligned tool-edge scratches are more prominent and extensive on the surface of the crown.

preparation of the margins of the lithic tools before retouching it [cf. 66]. This morphology of the marks suggests that this tooth was probably also used as a pressure flaker to shape or resharpen a tool edge.

## Bones used as knapping tools

Seven horse postcranial bones from Gough's Cave exhibit modifications consistent with their use as knapping tools. These include five distal metapodials (three metacarpal III and two metatarsal III), a complete metatarsal, and a proximal phalanx. A red deer distal metatarsal also has percussion damage consistent with its use as a knapping tool (Table 1).

**Metacarpal, *Equus ferus* (M49934).** M49934 is a distal end of a horse metacarpal recovered by Parry from spit 14 in November 1927 (Fig 4). Oblique cut marks are located on the medial and palmar surfaces on the diaphysis and a second set of shorter cuts is located on the articular surface (Fig 4B). Parkin *et al*. [46] recorded this grouping of cuts as a common feature of many of the horse metapodials from the Gough's Cave relating to one of the primary butchery tasks undertaken at the cave, which involved the processing of horse limbs to remove the long tendons.

Impact marks show that the bone was struck with heavy blows targeting the middle part of the shaft on the convex dorsal face. Patches of concreted sediment have penetrated the outer surface of the bone and it was not possible to remove these using mechanical and chemical

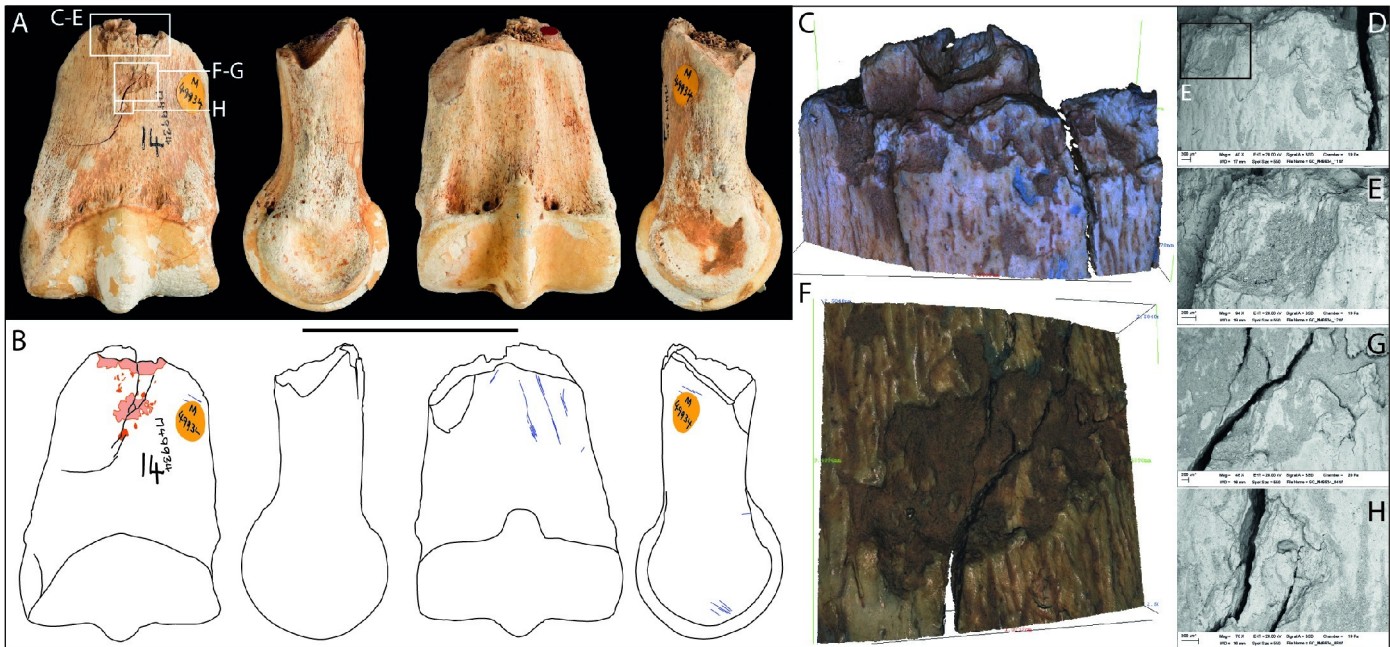

**Fig 4.** Photographs (**A**) and drawings (**B**) of horse distal metacarpal M49934, showing knapping damage (pink), cut marks (blue), and post-excavation flaking (light grey) in dorsal, medial, palmar, and lateral views (Scale = 50 mm). SEM micrographs and Alicona images detailing: (**C-E**) impact area associated with ancient transverse breaks and flaking with enlargements of the same; (**F-H**) cluster of large knapping pits and SEM enlargements showing internal features (partly obscured by cave-earth). Cracks are post-excavation features, possibly following lines of weakness created during knapping.

methods of preparation without damaging the bone surfaces. Although sediment partly obscures areas of the features, the areas of visible microtopography contain enough detail in the higher magnification images to show that these are knapping marks, possibly resulting from several superimposed blows. The deeper pitting at the proximal end of the specimen appears to have weakened the shaft (Fig 4C and 4F), which may have shattered with the final blow. The break profile associated with this break includes curved and angular fracture surfaces, a type of break commonly associated with breakage of fresh bones with a thick cortex [79].

**Metacarpal, *Equus ferus* (M50024).** Specimen M50024 is a distal fragment of a horse right metacarpal recovered by Parry from spit 18 in his 1927 excavation area (Fig 5A and 5B. See Table 2 for radiocarbon dates). Apart from some small areas of post-excavation flaking on the epiphysis and root marks on the diaphysis, the surface is well preserved. The diaphysis is broken obliquely with irregular angular and curved breaks and incipient cracks extending into the diaphysis. Sets of short transverse and oblique cut marks on the medial and lateral faces on the distal epiphysis and on the palmar surface of the shaft are from dismemberment of the forefoot and removal of the tendons. Similar cut marks are found on most of the horse metapodials from Gough's Cave, as noted by Parkin *et al.* [46]; this specimen is figured in their Plate 14, which draws attention to percussion damage where the bone was smashed.

Angular knapping pits and furrows are located on three faces, the marks being concentrated on the convex part of the lateral and medial faces and in the middle of the diaphysis on the dorsal face (Fig 5B). The angular pits are best illustrated by examples on the medial face (Fig 5I and 5J), which show features consistent with forceful contact with a sharp stone. A large (7 mm diameter) pit on the dorsal face could have been inflicted by a single blow (Fig 5C and 5D). This feature retains a remnant flake of displaced bone debris pushed up on one side of the

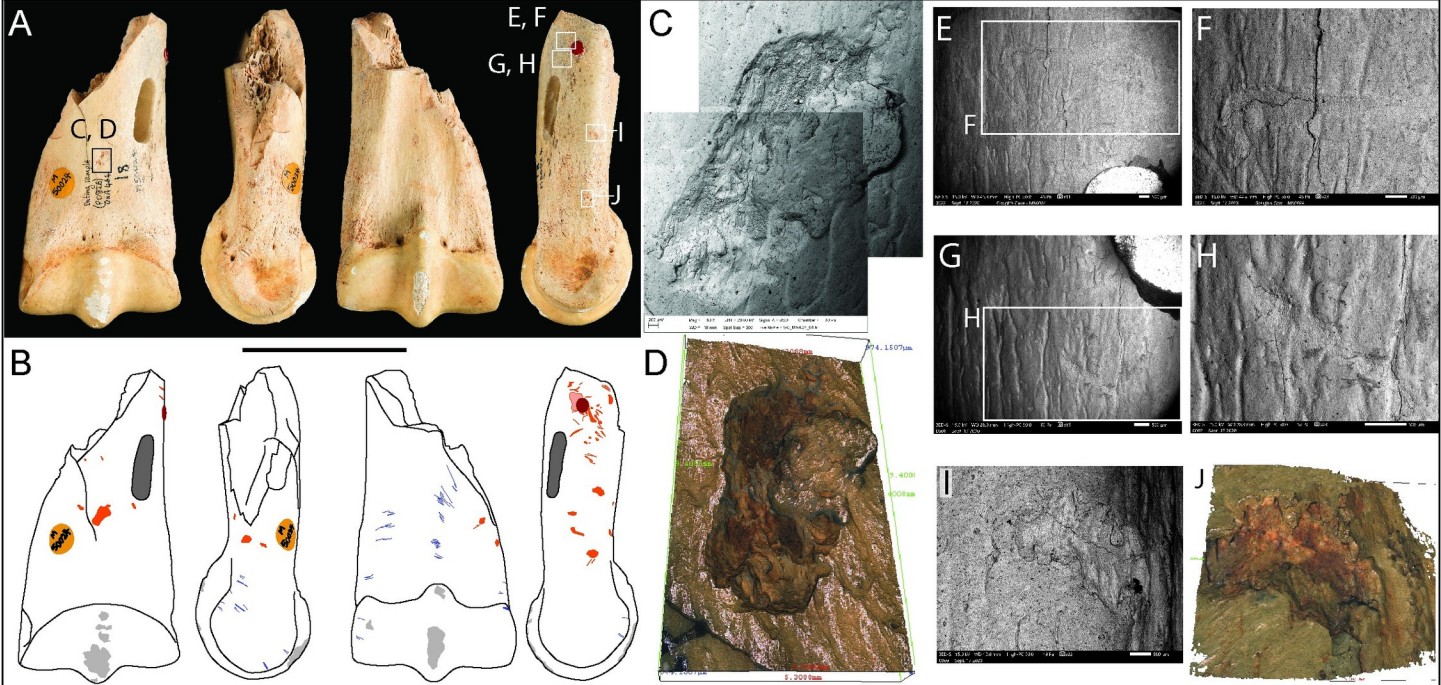

**Fig 5.** Photographs (**A**) and drawings (**B**) of horse metacarpal M50024, showing knapping damage (pink), cut marks (blue), sample hole (dark grey) and post-excavation flaking (light grey) in dorsal, lateral, palmar, and medial views (Scale = 50 mm). SEM micrographs and Alicona images detailing: (**C-D**) remnant of displaced bone adjacent to the pit; (**E-H**) scores; (**I-J**) knapping pit with an irregular profile from the chipping-away of bone matrix.

knapping pit; this is best seen in the oblique 3D image (Fig 5D). The furrows are superficial and, when examined using the SEM, appear to be featureless and lack striations often (but not always) associated with knapping damage (Fig 5E–5H). This tool shows only a few knapping features suggesting that it broke or was discarded after only a short period of use.

**Metatarsal, *Equus ferus* (M50000).** This fragment of a right metatarsal, which consists of the distal epiphysis and a portion of the shaft, comes from spit 15 of Parry's 1927 campaign. The diaphysis is broken across the lower third of the shaft with stepped and curved fracture faces (Fig 6A and 6B). Cracking affecting the distal epiphysis is recent, possibly from desiccation during storage. Cut marks on the palmar surface of the shaft match those observed in the previous specimen (metacarpal M50024) and are interpreted as cutting to detach the long tendons.

The knapping damage is conspicuous with a zone of concentrated and dispersed angular pits and gouges extending ~50 mm along the dorsal surface of the shaft (Fig 6A–6D). The macro-photograph, SEM and Alicona images (Fig 6E–6K) show that the dominant types of mark are gouges or elongated pits, most of which follow a similar, slightly oblique orientation to the axis of the bone. The SEM images show internal transverse micro-striations from contact of the knapping tool with irregularities on the edge of the stone tool being worked. It is evident that the knapping marks were present on the metapodial before it was broken because some of them are truncated by an ancient break surface. As with metapodial M49934, the break appears to have been initiated from an impact coinciding with other battering marks at the apex of the shaft on the dorsal face. Emanating from this knapping area is an incipient crack that extends distally along the shaft (Fig 6C).

The location and morphology of the knapping marks suggest that the shaft was held near its proximal end and swung during knapping with a forceful action, with the distal part of the

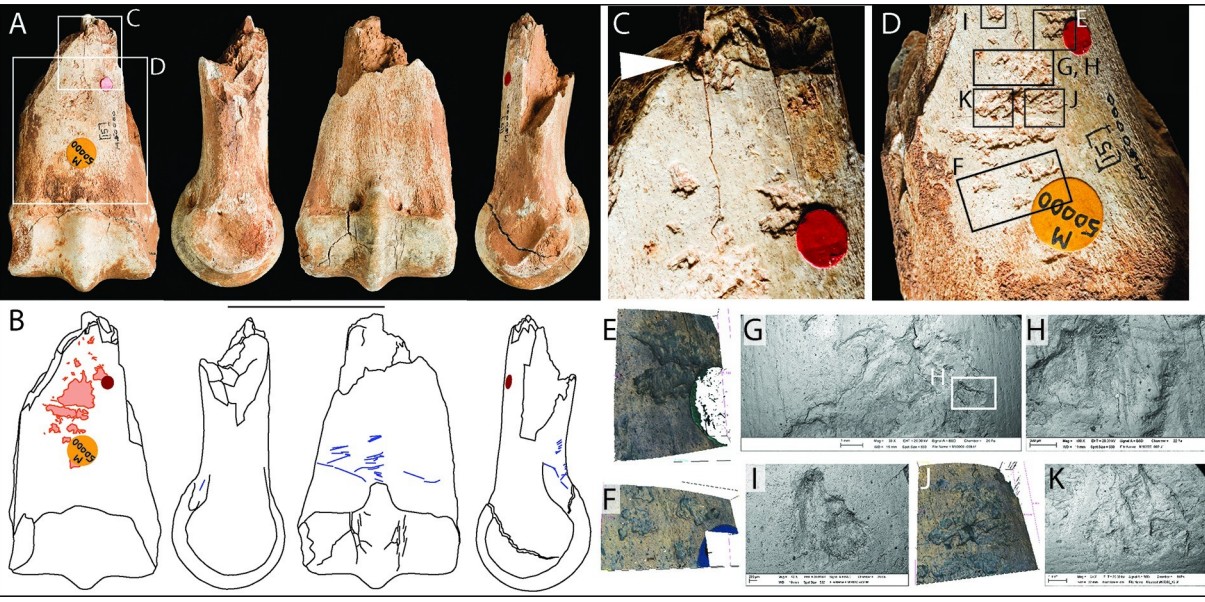

**Fig 6.** Photographs (**A**) and drawings (**B**) of horse distal metacarpal M50000, showing knapping marks (pink) and cut marks (blue) in dorsal, medial, palmar, and lateral views (Scale = 50 mm). SEM micrographs and Alicona images detailing: (**C**) close-up of impact (white arrow) at the apex of the shaft on the dorsal face and incipient crack that extends distally along the shaft; (**D**) close-up of knapping damage with areas of isolated and overlapping scores and pits figured in images **E-K**.

shaft making contact with the core or flake. This percussor was probably used during the initial working of a flint tool or detaching a blade from a core, rather than delicate removals required to shape a tool-edge.

**Metatarsal,** *Equus ferus* **(M49873).** M49873 is a complete metatarsal, which is unusual because it is one of the few intact horse limb-bones in an assemblage which otherwise shows a high incidence of breakage from marrow processing and tool manufacture and use [41]. It was found by Parry in 1927 (spit 13) and was sampled for radiocarbon dating by the British Museum Radiocarbon Laboratory (Table 2, Fig 7). The surface is well preserved, although areas are masked by concreted cave sediment which has partly filled some of the surface features. Unusually for a horse bone from Gough's Cave, no cut marks or other butchery traces have been observed on this specimen.

Knapping marks occur on several locations along the middle two-thirds of the shaft, with most of the marks being located on the dorsal and lateral and medial surfaces (Fig 7B). These are predominantly elongated pit and wedge-shaped gouges (Fig 7C–7F), some of which have cracks and displaced 'shoulder' of bone forming part of the impact feature (Fig 7C). Most of the more conspicuous features are located towards the distal end of the shaft, whereas the shallower linear depressions occur mainly in the mid-shaft region. The distribution of the marks suggests that during knapping, the bone was held near the proximal end and blows were made using the distal part of the shaft to remove larger flakes, whereas lighter blows utilised the middle part of the shaft, leaving shallow scores and 'pock marks'.

**Metatarsal,** *Equus ferus* **(M50025).** This fragment of a horse distal metatarsal from the 1927 excavation (spit 18) is stained with black patches of manganese oxide deposits and heavily marked by carnivore chewing. The carnivore damage is extensive on the distal end and shaft, and consists of furrows, scores and pits (Fig 8A–8D), probably created by a medium-size carnivore, possibly wolf or dog. Other surface features include cut marks, knapping marks, cracks and occasional root marks (Fig 8E and 8G).

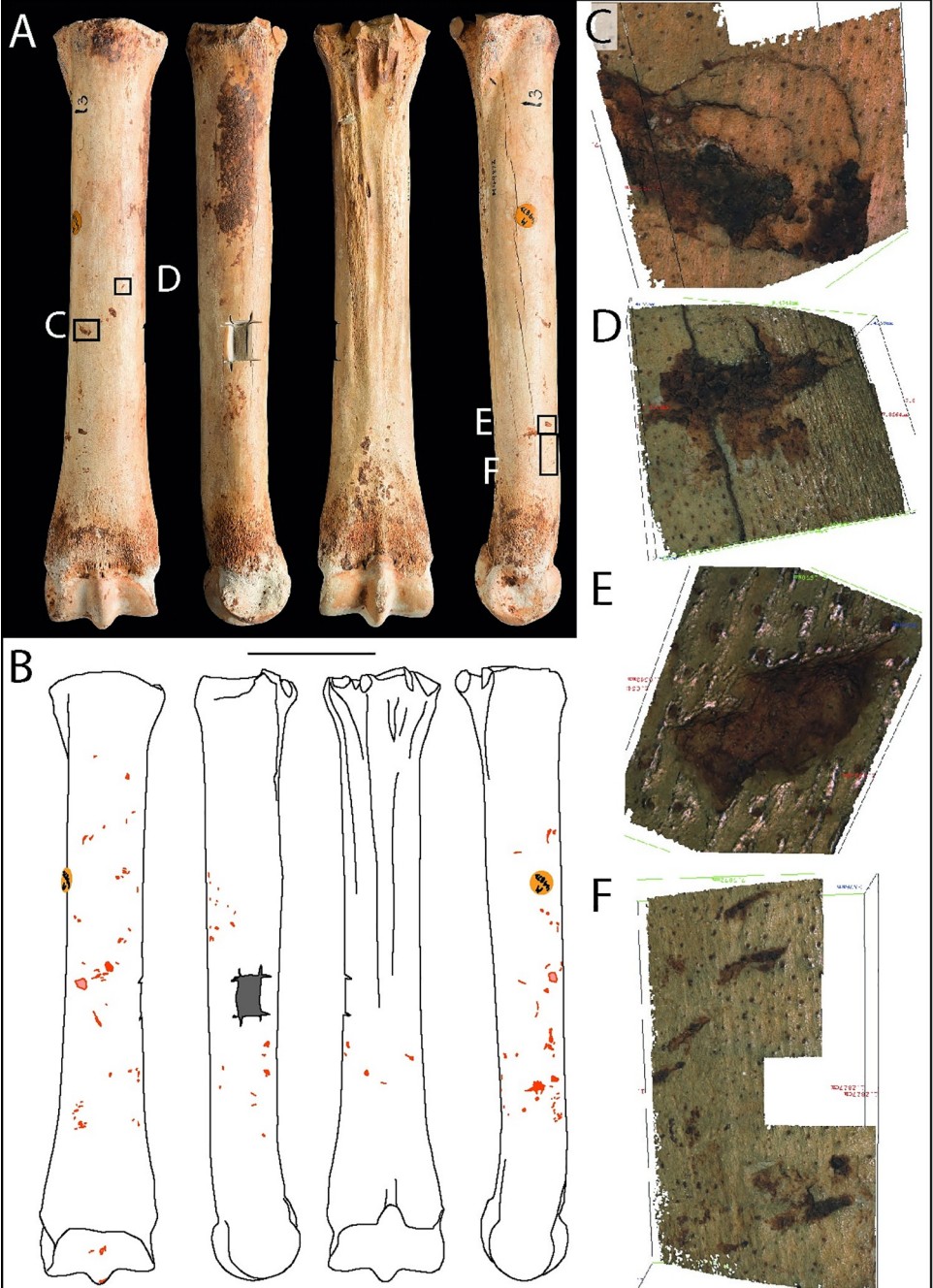

**Fig 7.** Photographs (**A**) and drawings (**B**) of horse metatarsal M49873, showing knapping marks (pink) and rectangular sample hole (dark grey) in dorsal, medial, plantar, and lateral views (Scale = 50 mm). Alicona images detailing wedge-shaped gouges (**C-E**) with concentric cracks demarcating depressed cortical bone (**C**), and shallower superficial scores (**F**).

The identification of this specimen as a knapping tool is based on the recognition that not all of the pits were created by carnivore chewing. Hidden among the conspicuous traces of chewing is a second set of pit-like features on the dorsal surface. These are compression features and gouges, mostly located away from the main concentration of unambiguous carnivore

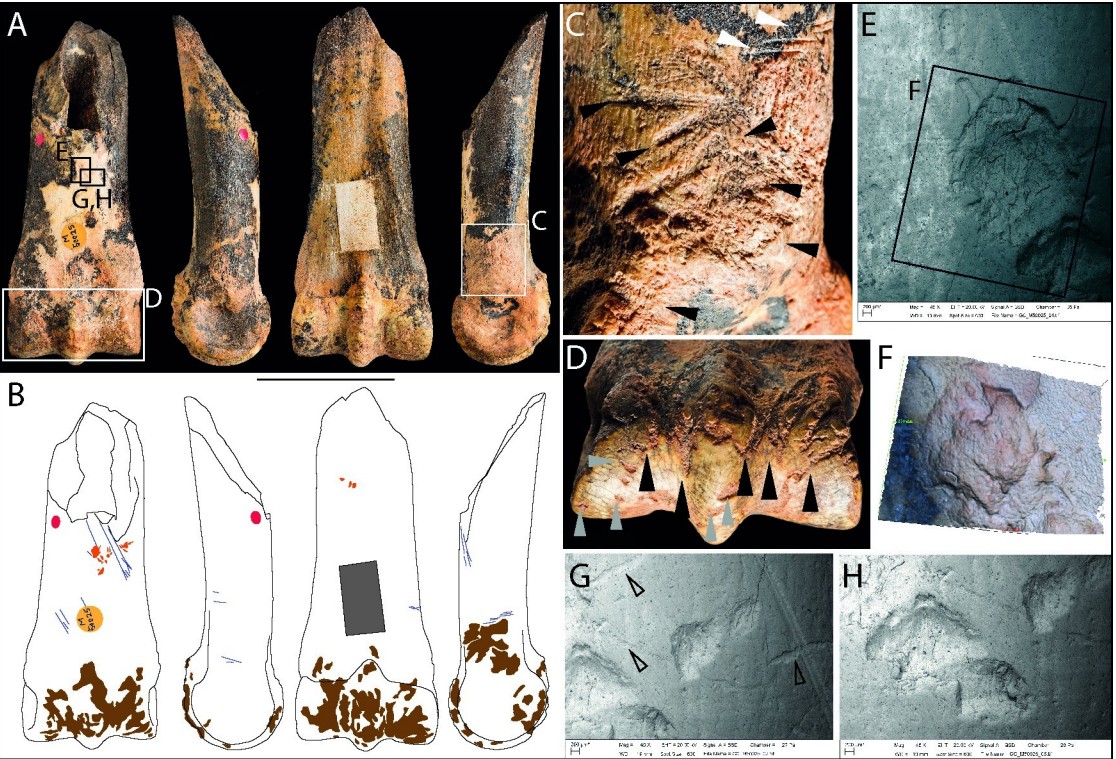

**Fig 8.** Photographs (**A**) and drawings (**B**) of horse metatarsal M50025, showing knapping marks (pink), cut marks (blue), carnivore chewing marks (brown) and sample hole filled with plaster of Paris (dark grey) in dorsal, medial, plantar. and lateral views (Scale = 50 mm). Cut marks (white arrows) and carnivore chewing marks clustering at the distal end (**C-D**) include furrows (black arrows) and pits (grey arrows). The gouged morphology of the knapping marks (**E-H**) distinguishing them from the compressed features of the carnivore tooth pits. Note the superficial root-etched channels in **E** and **G** (open arrows); these indicate the bone was buried in the daylight zone of the cave.

chewing marks (Fig 8E, 8G and 8H). The primary means of distinguishing the pits created by carnivore chewing from those resulting from knapping are morphological features, some of which are visible only with magnification (Fig 8).

Experimental studies have shown that the shape and microscopic features of knapping marks depends on the lithic material being worked and the type of knapping action (e.g. pressure- flaking or heavy blows; e.g. [77]). Perhaps the most significant factor, however, is the morphology of the lithic edge that makes contact with the knapping tool. The pits on M50025 include several examples with a highly distinctive 'mushroom-shaped' morphology (Fig 8C and 8D). The significance of these marks is highlighted in the Discussion.

The spiral break across the diaphysis is unlikely to have been caused by canid chewing; the dental apparatus of these medium-sized carnivores is simply incapable of cracking the extremely thick cortical walls of horse metapodials [79]. The breakage was more likely due to marrow extraction or the result of damage during the use of the metatarsal as a percussor.

**Metatarsal, *Cervus elaphus* (M49847).** This distal part of a left metatarsal from a red deer was recovered from spit 13 during Parry's 1927 season. Other than some minor excavation damage on the condyles and the removal of two bone samples (one drilled from the shaft, the other cut from the broken end), the surface of the specimen is well preserved (Fig 9A and 9B). AMS radiocarbon dates obtained from the samples place it with Magdalenian occupation (Table 2).

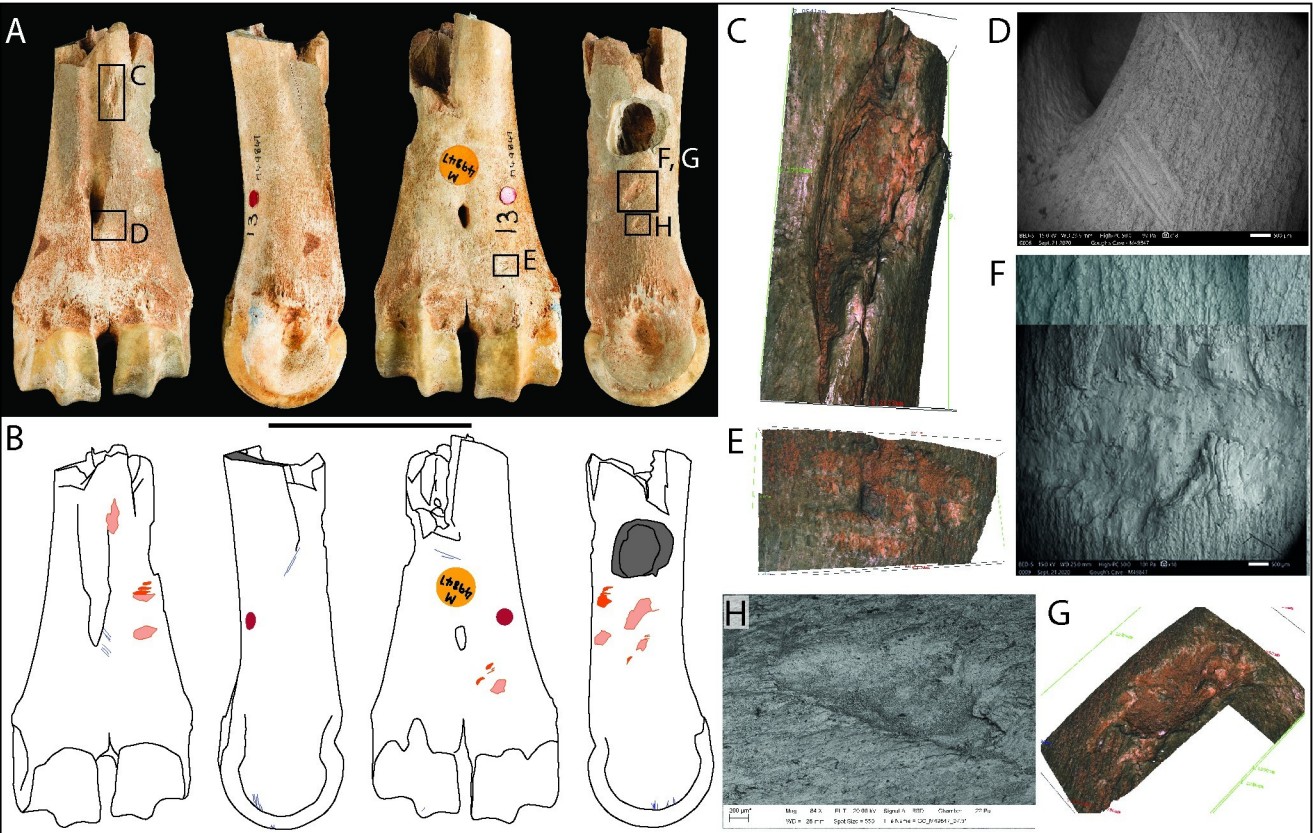

**Fig 9.** Photographs (**A**) and drawings (**B**) of red deer distal metatarsal M49847, showing knapping marks (pink), cut marks (blue) and samples (drilled hole = dark grey, cut piece = red oval) in dorsal, medial, plantar, and lateral views (Scale = 50 mm). SEM micrographs and Alicona images detail a point of impact with associated flaking (**C**), superficial slicing cut marks made with the tool held at an oblique angle to the bone surface (**D**), and knapping marks on the plantar and lateral faces (**E-G**).

Cut marks on either side of the epicondyles are from disarticulation of the foot, and the longer oblique cut marks on the shaft were probably inflicted during skinning and tendon removal (Fig 9D). The bone was broken when fresh, with 'green' breaks resembling those of other thick-walled limb-bones from the site that were broken to extract marrow and the horse metapodials that were broken during use as knapping hammers. This specimen is considered to be a flint-knapping hammer from the impact features on the distal end of the shaft. These include angular pits with flaked margins and three knapping areas on either side of the central groove on the dorsal face (Fig 9C and 9E–9H), which are associated with larger areas of flaking, one of which has detached a flake of bone measuring ~20 mm (Fig 9C).

**Proximal phalanx,** *Equus ferus* **(UNREG 3482).** This proximal phalanx from a horse is complete and microscopic surface features are well preserved (Fig 10A and 10B). It was recovered from an exploratory excavation adjacent to BM(NH) Area I in 1987 (find number 23). This is arguably the best example of a retoucher from Gough's Cave, with comparable examples known from Magdalenian sites on the European mainland (see Discussion) as well as an earlier Mousterian example [80].

The phalanx is marked by numerous cut marks (Fig 10B and 10H). These were created during the severing of the complex set of ligaments that attach to the phalanges, and from cutting between the phalanx and metapodial to disarticulate this part of the foot [cf. 46].

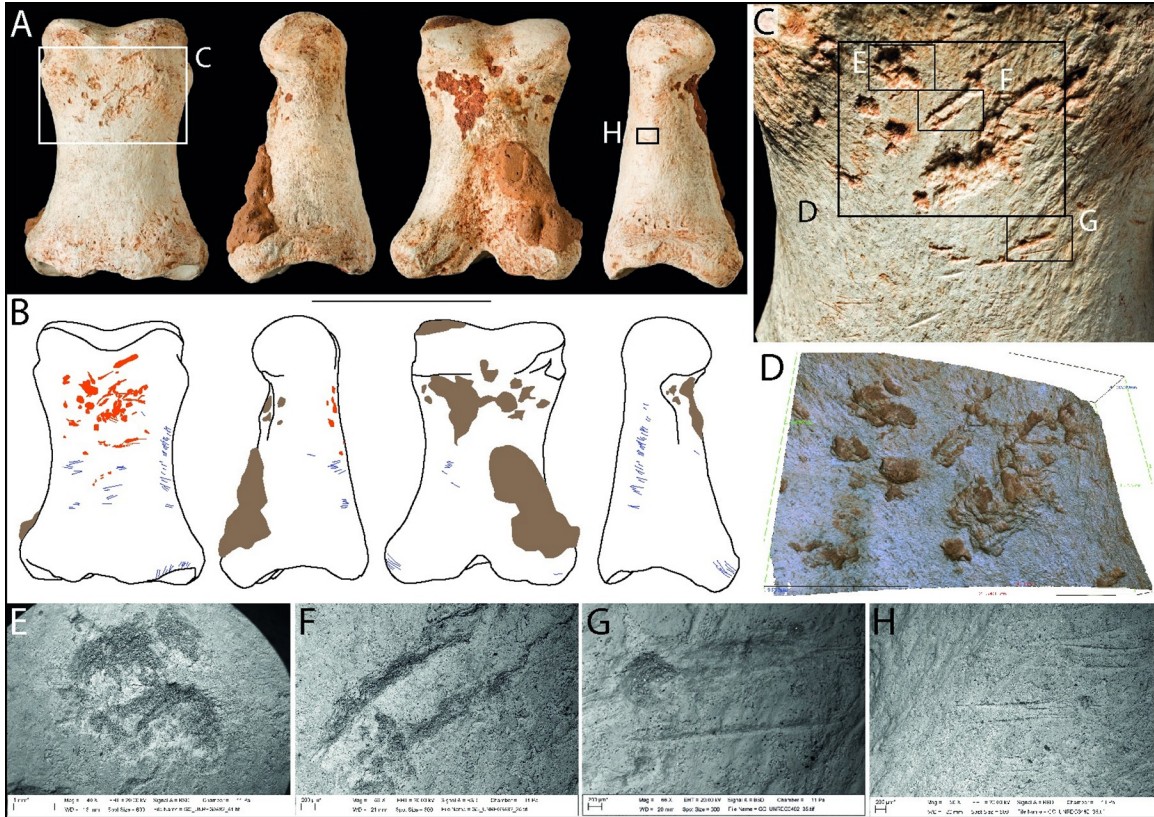

**Fig 10.** Photographs (**A**) and drawings (**B**) of horse phalanx I Unreg. 3482, showing knapping marks (pink), cut marks (blue) and concreted sediment (brown) in dorsal, medial, plantar, and lateral views (Scale = 50 mm). Close-up of knapping area (**C**) and Alicona (**D**) and SEM micrographs (**E-G**) of knapping pits and scores on the dorsal surface and cut marks on the side of the bone (**H**).

The knapping damage is concentrated on the flatter part of the dorsal surface of the shaft, close to the distal end (Fig 10A and 10B). The knapping marks include angular pits and linear grooves with an oblique orientation across the long axis of the phalanx (Fig 10C–10G). The profiles of the groves are angled (having one side steeper than the other) and microscopic examination reveals internal micro-striations orthogonal to their long axes; some of these striations are also visible in the macro-photograph (box 'F' in Fig 10C), but they are less clearly defined in the SEM images (Fig 10F). The grooves have a wavy plan-form that appears to have been created by impact against an irregular (or 'serrated') edge of a tool.

The knapping marks are consistent with the bone having been used in direct percussion rather than pressure flaking. Examination of cross-sections of the gouges indicates that during knapping, the phalanx was held at the proximal end. Moreover, the size of the bone allows it to fit snugly in the palm of the hand. The linear plan-form and relatively shallow depth of the marks, suggest that this tool was probably used to retouch the edges of flint tools.

The knapping tool assemblage is summarized in Fig 11, which shows outlines of the complete bones and areas of knapping damage.

## Interpretation and discussion

What can the knapping tools tell us about activities undertaken in the cave, and how were these tools integrated into the Magdalenian flintworking technology?

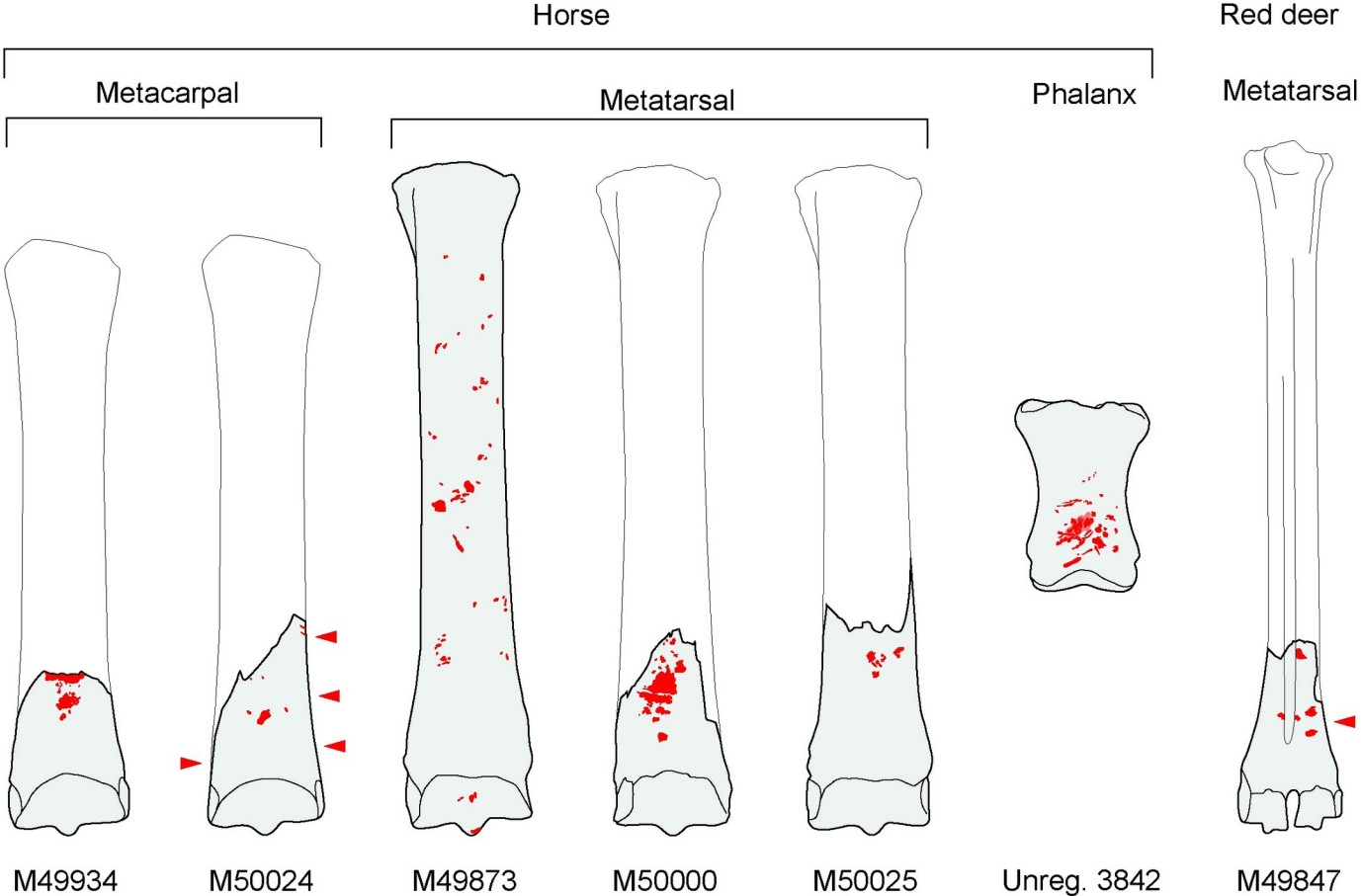

**Fig 11. Schematic outlines of postcranial bones from Gough's Cave used as knapping tools.** Red indicates knapping damage; arrows mark the location of impact features on the side of the bone. Note the consistent pattern of breakage of the metapodial shafts close to the distal articular end.

### Spatial analysis and activity areas at Gough's Cave

Two aspects of the spatial distribution of the knapping tools were examined using records of the horizontal and vertical distribution of the finds. The first part of this analysis examined the spatial distribution of the knapping tools within the cave. This is possible because earlier excavations at Gough's Cave recorded finds according to excavation areas, and vertically from an arbitrary datum (Figs 12 and 13). Specimens examined for this study can usually be assigned to a 'zone' within the cave, even when detailed information on the circumstances of the find has been lost. With these limitations in mind, studies of the faunal remains by Parkin *et al.* [46] and Currant [41] identified the spatial limit of the faunal remains and artefacts in the cave and established that both flints and bones increase in abundance towards the daylight zone. Fewer artefacts were found in the Vestibule suggesting that little regular human activity is likely to have taken place this far back in the cave. The higher proportion in the Vestibule of carnivore gnawed bones (including specimens with cut marks) suggests that dogs and foxes carried bones originally discarded by the human occupants to their feeding refuges at the back of the cave [46].

Deposits at the front of the cave and the slope from the cave mouth into the gorge were removed during the early stages of the development of the site as a tourist attraction. These deposits were largely cleared during subsequent alterations to the gates and the construction,

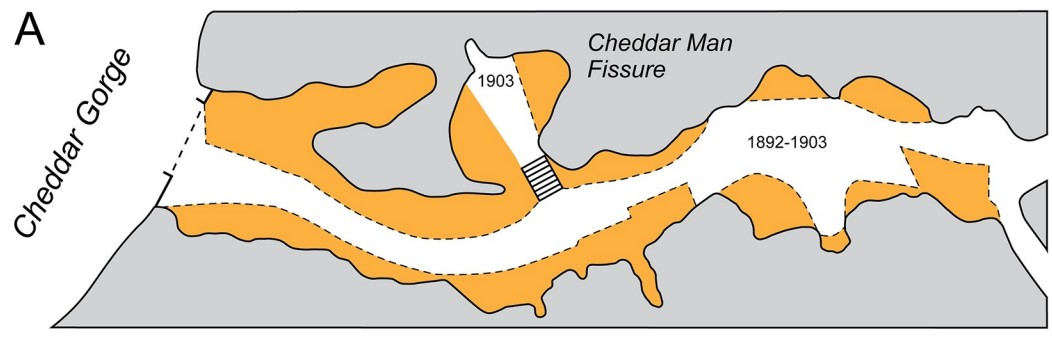

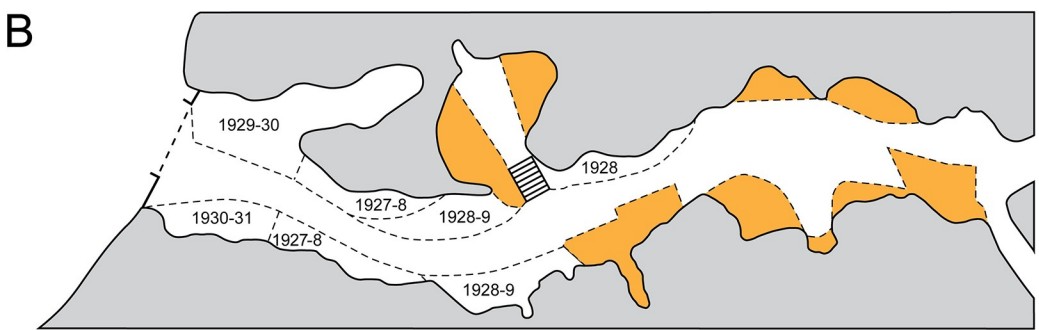

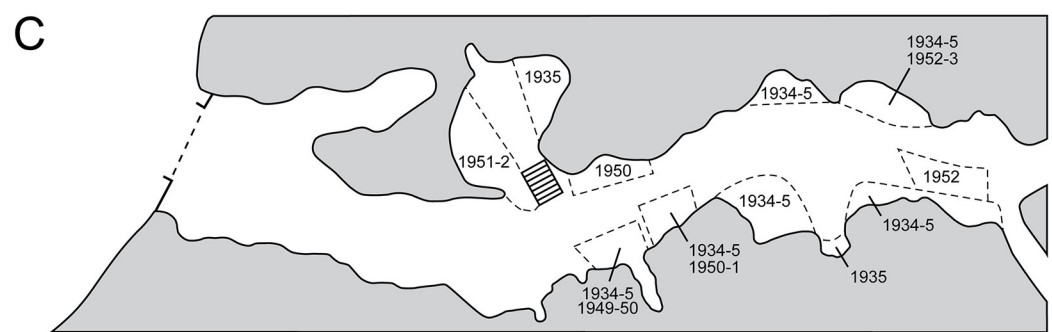

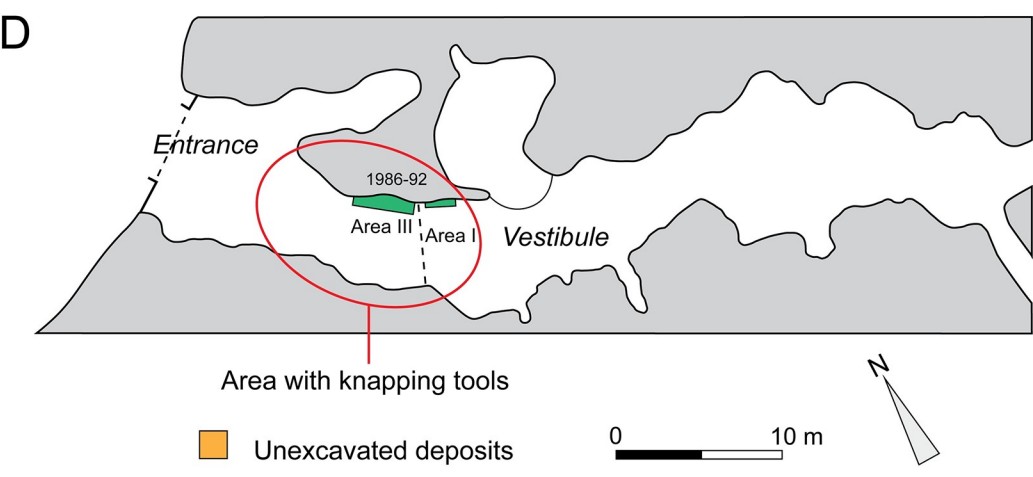

Area with knapping tools

Unexcavated deposits

0    10 m

**Fig 12.** Plans of the outer part of Gough's Cave showing stages in the removal of the Pleistocene sediments: **A** 1892–1903; **B**, 1927–1931; **C**, 1934–1952; **D**, 1986–1992 with dashed line indicating the grill gates that separate the Entrance Chamber from the Vestibule (plans based on [13–15]). Area with knapping tools bounded by the red oval.

and alterations of the visitor's centre. Between 1892 and 1903, a narrow footpath was cut through deposits to allow access for visitors to the deeper recesses of the cave and a large side chamber (Cheddar Man Fissure) was cleared by the cave management (Fig 12A). The most display-worthy artefacts and animals bones were kept, but only a few of these display-items can be located today. Starting 24 years later and continuing until 1931, R.F. Parry coordinated excavations at the site, producing a wealth of faunal remains and artefacts. Parry excavated a series of trenches on either side of the pathway by removing the cave sediments in six-inch (c. 15cm) layers (spits) numbered from top to base, and to the level of the pre-existing pathway (Fig 13). Over this five-year period, the excavation progressed systematically from the entrance gate for a distance of approximately 25 m to the mouth of the Vestibule (a small area within the Vestibule was excavated in 1928) (Fig 12B). Nearly all the remaining fossiliferous sediments were removed in 1934–35 and 1948–53, during operations in two episodes undertaken by the cave management to further improve public access. These excavations were located at the back of the cave, within the darker recess of the Cheddar Man Fissure and in smaller areas along the walls of the main chamber up to 50 m from the mouth of the cave (Fig 12C). The most recent phase of excavation took place in 1986–92 when two smaller trenches were opened, one in the Entrance and the other in the Vestibule (Fig 12D).

The bone assemblage examined for knapping tools come from Parry's 1927–1931 excavation and the NHM excavation in 1986–92. Bones in other collections (Cheddar Caves Museum, Wells Museum, Taunton Museum) and those recovered from site works between 1934 and 1953 were unavailable for analysis during the Covid-19 pandemic. Parry's finds are associated with useful contextual information, namely the layer (spit) number and year of recovery, which are marked on the bones in pencil and ink. It is this information that allows attribution of the finds to narrow strips (~2 x 10 m) of previously unexcavated sediment that

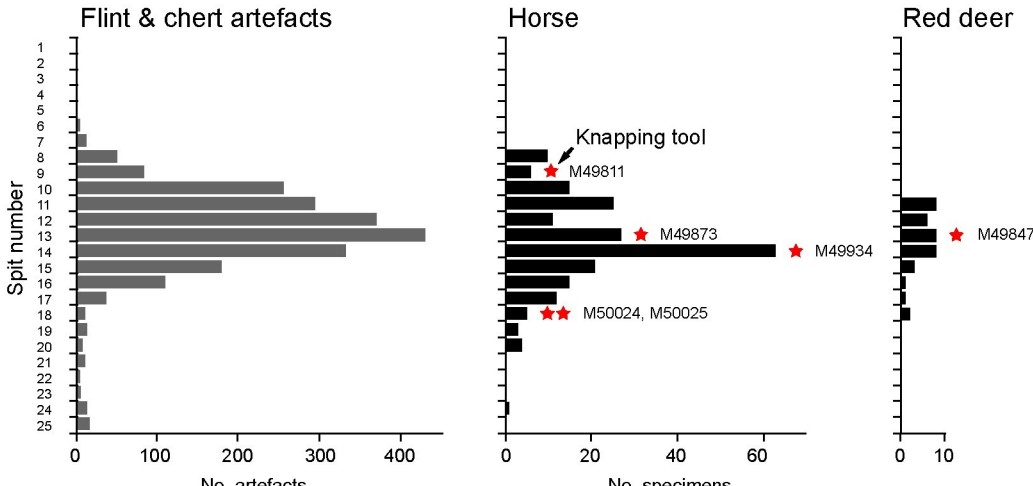

**Fig 13. Gough's Cave: Vertical distribution of flint and chert artefacts (based on [13]), Pleistocene horse and red deer bones [41, Table 1] and knapping tools recovered during excavations supervised by Parry in 1927–1931 (Fig 1).** The deposits were excavated in six-inch (~15 cm) layers (or spits). Note that pottery and Holocene bones are confined to spits 1–9 and the horizontal spits cutting through the natural slope of the deposits gives the illusion of stratified sequence [13, 41].

flanked the access path. Finds from the 1986–92 excavations were accurately surveyed to allow a more precise plotting of pieces. Although limited to two small trenches, the total recovery of finds contributes to understanding the coarser-grained spatial distribution of finds recorded by Parry.

First, the spit information recorded by Parry can be used to plot the vertical distribution of the knapping tools against those of the Palaeolithic lithic artefacts and the Pleistocene horse and red deer bones (Fig 13). The plot shows that although the knapping tools are dispersed vertically by as much as 0.5 m, they fall within the general distribution of Magdalenian artefacts and Pleistocene bones as recorded by spits. Three knapping tools (complete horse metatarsal M49873, horse distal metacarpal M49934, and red deer distal metatarsal M49847) are from spits that yielded the highest number of lithic artefacts. At this level, Parry [19, p.46; 20, p.104] recorded '. . .a band of very black earth with charcoal and burnt bones. . . not continuous, but very distinct in places. . .'. The other knapping tools come from the areas near the top (horse incisor M49811) and bottom (horse distal metapodials M50024 and M50025) of the general dispersal of Upper Palaeolithic finds. Currant [41] and Jacobi [13] warn against accepting at face value the depth information to imply a pattern of an initial occupation by a pioneering group, followed by a longer period of intensive occupation, which saw a gradual decline as local environmental conditions became disadvantageous for horse hunting. Instead, they argue convincingly that this apparent vertical disposal through as much as 3 m sediment is an artefact created by the excavations methods with horizontal spits cutting across the natural stratigraphy of the sediment cone. The fossiliferous sediment exhibited a complex geometry as they originally formed a wedge sloping from the front of the cave to the interior, and steeper slopes laterally towards the cave walls. This is borne out by Jacobi's [13] discovery of refitting flints that cross several spits as well as by the statistically indistinguishable radiocarbon dates (incorporating dates on two knapping tools) that are dispersed between spits 5 to 24. From this, they conclude that there was a single Magdalenian occupation horizon that was deposited on a sloping surface.

The second part of the spatial analysis was to determine whether the knapping tools are scattered in an isolated fashion in different parts of the cave or concentrated in areas with knapping waste. This was addressed using the information (year of excavation) which allows finds to be assigned to a 'zone' within the cave (Table 2, Fig 12). The nine knapping tools from Parry's excavations were all donated to the BM(NH) in 1928 and of these, five were recovered in November 1927 and two were found during the 1927 campaign. The details for the remaining two specimens are less precise, but suggest they were found either in 1927 or during the 1928 season. The single knapping tool from the BM(NH) excavation was recovered from Area I, just inside the Vestibule and immediately adjacent to an area excavated in 1927–8. This part of the cave is within the daylight zone, which accounts for the fact that three of the metapodial knapping hammers exhibit root etching (Table 3) implying that the bones were exposed before burial in an area that had sufficient light for vegetation to grow.

Although the number of knapping tools currently known from Gough's Cave is small relative to the total number of bones from the cave, they appear to be concentrated in a small area within the rear part of the Entrance Chamber (Figs 1 and 12), between about 10 and 20 m from the mouth of the cave. The suggestion that this is an area of the cave where flint knapping took place is supported by information from the early excavations and Jacobi's [13] in-depth analysis of the flint artefacts. The most compelling evidence that knapping took place in this part of the cave comes from observations by H.N. Davies [81, p.339], who described a rectangular block of limestone toward the centre of the Vestibule at the base of which were found '. . . a large number of flint-chips embedded in the earth. . .'. More chips were on its upper surface. Davies believed that this '. . . tabular block had apparently served as a tool-bench to some cave-

**Table 3. Gough's Cave knapping tools.** Summary of taphonomic alterations.

| Museum no. | Anatomical element | Portion | Cut mark | Knapping damage | Carnivore gnawing | Root marks | Modifications after excavation |
|---|---|---|---|---|---|---|---|
| NHM PV M50064 | M$^3$ | C | | P, S, Fl | | | |
| NHM PV M49811 | I$^1$ | C | | P, S, TES | | | |
| NHM PV M49934 | Metacarpal | DS | Y | P | | | Fk, Cr |
| NHM PV M50024 | Metacarpal | DS | Y | P | | Y | Fk, S |
| NHM PV M50000 | Metatarsal | DS | Y | P, S | | | Cr |
| NHM PV M49873 | Metatarsal | C | | P, S | | Y | Cr, S |
| NHM PV M50025 | Metatarsal | DS | Y | P | Y | Y | S |
| NHM PV M49847 | Metatarsal | DS | Y | P | | | S |
| NHM PV Unreg. 3482 | Phalanx | C | Y | P, S | | | |

**Portion:** C = complete element, DS = distal end with portion of shaft. **Knapping damage:** P = pit, S = score, TES = tool edge scratches, Fl = embedded flint chip.
**Modifications after excavation:** S = sampled for radiocarbon dating or other analyses, Fk = surface flaking, Cr = desiccation cracks.

dwelling worker in flint. . .' (ibid.). Subsequently, Parry [20, p.103] observed that '. . . the flint cores and numerous chips seem to go to prove that the tools were manufactured on the spot. . .'. That knapping took place in the cave is supported by Jacobi's analysis of the surviving lithic assemblage. He identifies pieces removed from cores to facilitate blade removal (*flancs du nucléus*), core tablets and as many as 60 platform preparation flakes that provide direct evidence for flint knapping having taken place in the cave. Further confirmation comes from minute spalls and flint flakes recovered in the water-sieved residues from the 1986–92 excavation; these were found in the same area as partially refitting knapping debris including platform preparation flakes, some of which refit with each other [13, p.20].

These observations, coupled with the reasonable assumption that the distribution of knapping waste is concentrated in the area where the production of the blades took place and/or where tools were shaped and re-sharpened, suggest that the main knapping area(s) in the cave was situated towards the back of the Entrance Chamber and the front of the Vestibule (Fig 1B).

It may be significant that the knapping tools from this area include both curated and heavily used knapping tools (the horse phalanx and molar) that were probably utilised to shape, re-sharpen or reconfigure (retouch) tool edges, as well as the metapodial knapping hammers. The latter tools appear to have been dropped where they broke during use; the breakage probably resulted from the heavier blows required to detach blades from a core. Location of these activities in the cave may also be significant given that this area is located both within a sheltered part of the cave and in the daylight zone where tasks could have been undertaken without artificial light.

## Knapping tools and the lithic *chaîne opératoire* at Gough's Cave

Experimental knapping kits may include several different types of tool made of a variety of materials (stone, wood, metal, bone, tooth, antler) depending on the tool type and the cultural context and the artefact being replicated [82, 83]. Broadly, knapping tools can be assigned to one of the following categories:

1. Hard hammers: typically cobbles or pebbles of different sizes used as percussors to test and roughly shape a rock. Hard hammers may be the only knapping tool required to make crude handaxes and flake tools.

2. Soft hammers: soft stone, antler and bone percussors or wooden billets [e.g. 84] used to remove blanks (flakes or blades) from a core or to shape and re-sharpen stone tools (active

retouching, *sensu* Starkovitch *et al.* [85]). Hard and soft hammers are also used to abrade platforms; this is done to facilitate accurate flake-removal.

3. Punches: for removing flakes by indirect percussion.

4. Anvils: stones, large bones or wooden supports used for indirect flaking and bipolar knapping.

5. Pressure-flakers: Antler tines, pointed bones or teeth used to trim the edge of a stone tool by forcing-off small flakes, by mean of passive or active pressure flaking [75, 78, 85].

Palaeolithic archaeologists recognize a further category of knapping tool classified as retouchers. These can be complete bone, bone fragments, antlers and teeth used to shape, sharpen, re-sharpen, re-purpose or repair the edges of stone-tools. Starkovitch *et al.* [85] define two modes for tools used in retouching, namely 'active' in which the knapping tool is forced against the edge of a stone, and 'passive' in which the retoucher is stationary and the edge of the stone tool is pressure-flaked against it. Retouching can be accomplished with a variety of tools, including hard hammerstones, soft hammers (stone cor organic), punches and anvils as well as pressure-flakers.

Identifying and interpreting prehistoric knapping tools involve detailed observations of the marks and comparisons with damage on tools used in knapping experiments where the actions and knapping activities are known [e.g. 67, 68, 74, 75, 77, 78]. It is also possible that some of the superficial semi-parallel scratches were produced during preparation of the margins of the lithic tools before retouching it [cf. 66]. Using this approach, it has been possible to identify three distinct uses of the knapping tools in the Gough's Cave assemblage based on the micro-morphology of the knapping damage and the overall characteristics of the specimens (Fig 14):

a. Hammers/percussors: metapodials (M49847, M49873, M49934, M50000, M50024, M50025) were swung in the manner of a hammer using heavy blows to detach blade- and bladelet-blanks from a core (Fig 14A). An unusual set of 'mushroom-shaped' knapping marks on one of the metapodials (M50025) is probably related to a characteristically Magdalenian method of facetting the butts of blades, the creation of the so-called *talons en éperon*. This preparation technique allowed greater control over the angle between the striking platform and the core-face to guide the percussive blow and perhaps facilitate longer removals [13, 26]. The impact features appear to be the imprint of *talons en éperon*, the unusual form of which results from the small-scale convergent removals that define the spur on the core edge ([13], Fig 10, p.35, part 8–9).

b. Active retouchers: phalanx (UNREG. 3482) and metapodial (M49873) used in gentler actions with the tool tapped against a sinuous flint edge, during shaping or re-sharpening tasks (Fig 14B).

c. Pressure-flakers: teeth (incisor M49811, molar M50064) utilized as pressure flakers ('compressors') in delicate and precision tasks, such as backing edges, removing small chips to shape a cutting edge, or to re-sharpen a tool (Fig 14C and 14D).

The term 'retoucher' is generally applied to simple *ad hoc* tools that utilised discarded bones or bone fragments obtained from the butchery waste (e.g. long-bone shaft fragments for marrow processing). In this scenario, they are interpreted as having been selected from waste close at hand and discarded, after a short period of use, on the spot where they were deployed [86]. The Gough's Cave examples show that these tools can have more complex 'biographies' (Fig 15). The metapodials knapping tools from Gough's Cave almost certainly represent such

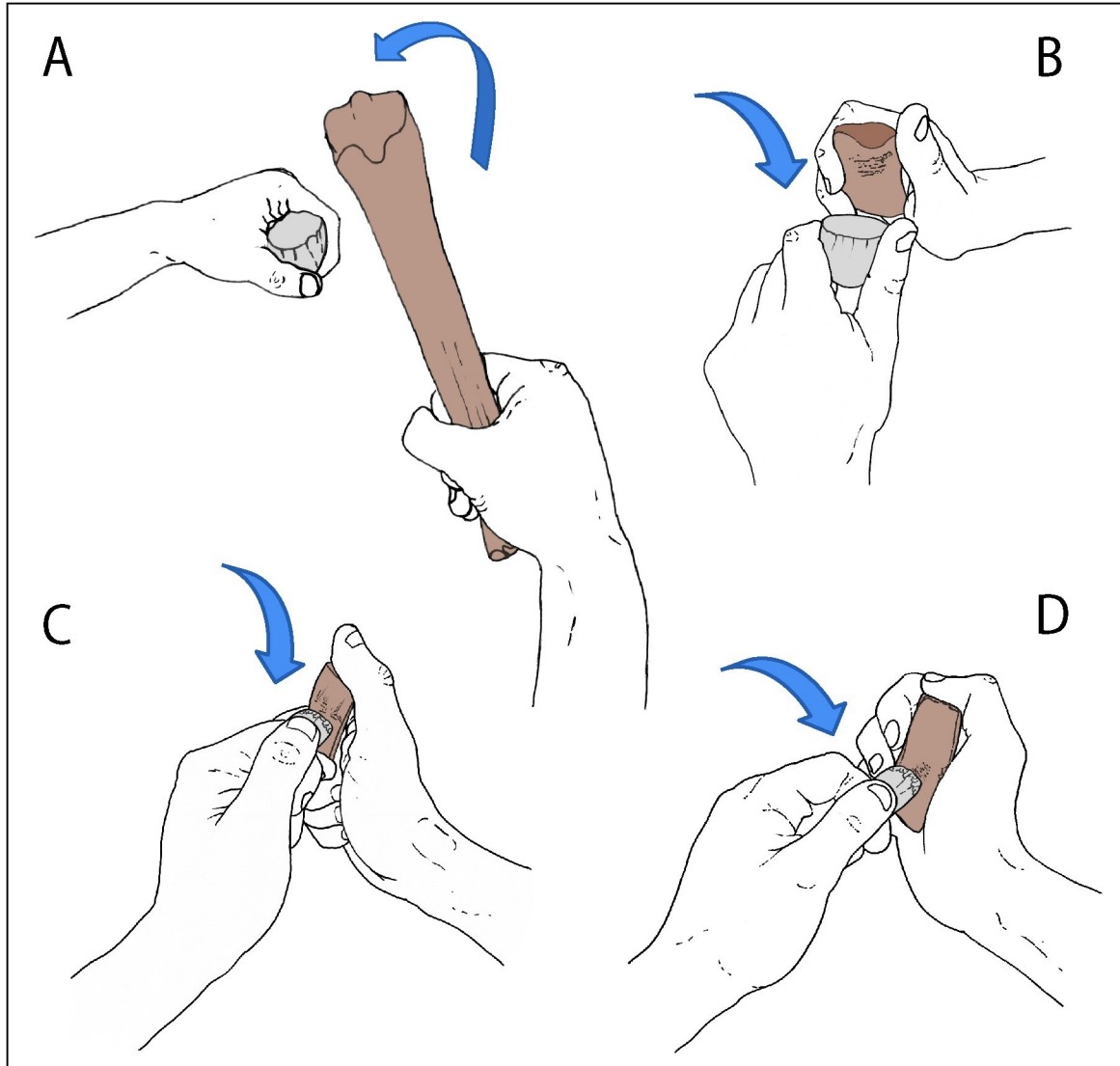

**Fig 14. A-D. Magdalenian techniques of knapping flint tools at Gough's Cave.** A. Horse metapodial (e.g. M50025) used as a hammer to detach blades/bladelets from a core; B. horse phalanx (UNREG 3482) used as a hammer to remove small blades or flakes, or for retouching an edge; C. horse incisor (M49811) used as an active retoucher (*sensu* [85]) in pressure-flaking; D. horse molar (M50064) used as a passive retoucher (*sensu* [85]) in pressure-flaking.

'recycled' bone waste [cf. 86] that provided a source of expendable knapping tools. Although the incisor and the phalanx retouchers, and the metapodial hammers from Gough's Cave were probably used to remove very few blades during short episodes of flint-working, the horse molar pressure-flaker was evidently a curated item that had been used to retouch flint tools over an extended period of time before being discarded or lost. The Gough's Cave organic knapping tools therefore represent both *ad hoc* and curated tools. A similar spectrum from simple *ad hoc* knapping tools to curated items is known at least as far back as the early Middle Pleistocene Acheulean site at Boxgrove (Sussex, UK). At Boxgrove, the knapping tools include heavily utilised antler knapping hammers [4] from the Waterhole Site, and at the Horse Butchery Site, an acetabulum was brought in with the flint raw material for use during the initial

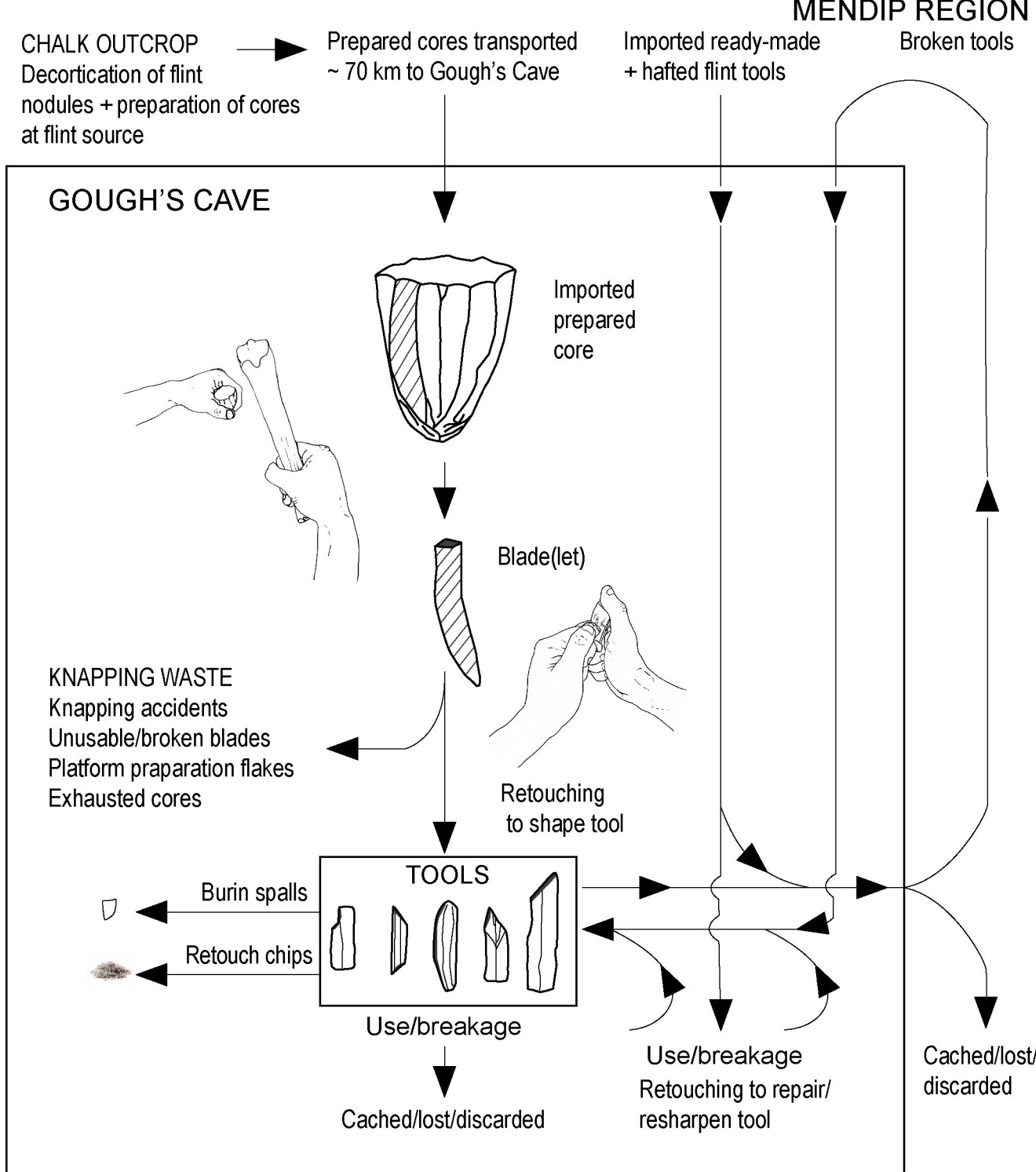

**Fig 15. Integrating the lithic *chaîne opératoire* with the knapping tools at Gough's Cave.**

stages of knapping, whereas bone shaft fragments (from breaking of the horse bones for marrow) were used to re-sharpen cutting edges as the butchery progressed [8].

The metapodial knapping tools from Gough's Cave were probably complete when they were used as knapping tools; the breakage likely occurred during the knapping process. This can be demonstrated for at least two of the metapodials (M49934, M50000) which show the deepest knapping marks located on the break. These impact features indicate forceful blows that may have been sufficient to shatter the bone. A consistent pattern of breakage is also exhibited by the other fractured metapodials (M49847, M50024, M50025), although none of these specimens have obvious knapping features bisected by the breaks.

These bone tools also display the highly characteristic set of cut marks indicating that they derive from butchery waste and the specialised processing of legs to extract the long tendons from the lower limbs [46]. A more intriguing example is the horse metatarsal M49873. Not only is this the only complete metapodial percussor (with relatively shallow knapping features), it also lacks indications (i.e. cut marks) that it was extracted from a butchered carcass. One possibility is that this bone was recovered from the decayed carcass of a horse that had died naturally and the metapodial was brought to the cave as a useful knapping tool. Although the use of 'dry' or 'semi-dry' bones has not been reported from other Upper Palaeolithic sites, there are earlier examples where weathered bones were used as knapping tools (e.g. Schöningen, Germany; [2]; and Lingjing, China [87]). There are also examples of defatted bones used as retouchers; these pieces were probably collected and used several months or years after being discarded as butchery waste (e.g. Prado-Vargas, Spain [88]; Orgnac 3, Cagny l'Epinette and La Grotte du Noisetier, [67, 89]).

Whether bone fragments were also used as knapping tools at Gough's Cave is more difficult to ascertain. This is because the bone collection from the 19th and early 20th century excavations are strongly biased towards specimens that are clearly identifiable, such as teeth, complete phalanges and metapodial epiphyses [41]. The missing component in this assemblage are featureless long-bone shaft fragments, which are known to have been used as retouchers at other Magdalenian sites (see below), as these seemingly uninformative specimens were kept following their excavation at these other sites. The Gough's Cave collection at the NHM also includes material that was collected during the 1988–1992 excavation. This collection was examined in detail as complete recovery of all faunal remains (including microfauna [44]) was ensured by the meticulous excavation techniques employed and the thorough wet-sieving of the excavated sediments. This is the source of the horse phalanx retoucher (GC 87 find no. 23 = UNREG. 3482), but careful examination of the shattered bones failed to identify any pieces with knapping damage. Similarly, our analysis of the human remains from Gough's Cave [31, 35] showed that, although nearly all the human remains have cut marks, marrow fracture impact features and even human tooth marks, none of these fragmented bones had been used as knapping tools. Although the Gough's Cave assemblage is far from complete, with a strong bias in some of the sub-samples, it would appear from our analysis of the 1988–1992 assemblage that long bone shaft fragments were not used regularly to work stone tools at the site.

None of the knapping tools show traces of preparatory working to make them more ergonomic or to improve their efficiency. This is in contrast with other assemblages, for example the late Middle Pleistocene examples from Schöningen [2], where the knapping areas on many of the bones have preparatory scraping to clean the bone of meat and periosteum. Other examples, such as the horse acetabulum from the Boxgrove Horse Butchery Site [8] or fragments of cave bear bones from Scladina cave [63], have been shaped to make them easier to handle. The use of complete bones and the absence of preparatory cleaning on the Gough's Cave bones suggest that the knappers did not consider this a necessary operation. Alternatively, the bones

were already in a clean condition with the soft tissue having been 'removed' from a joint that was roasted or boiled making it possible to clean the bone without having to use a stone tool. This supports the suggestion that the bones used as knapping tools at Gough's Cave were selected from butchery or cooking waste, and that these bones were chosen as they were already well-suited ergonomically/morphologically for this purpose.

Horse bones provide the bulk of the large mammal remains identified from Gough's Cave, whereas red deer bones provide a smaller proportion of those identified from the site. Accurate quantification of the Gough's Cave fauna has yet to be undertaken. Work on material from 1986–7 excavations is ongoing and the older collections are dispersed across several collections (Somerset Museum in Taunton, Cheddar Museum and NHM), not all of which have been published in their entirety. Nevertheless, quantification of sub-samples undertaken by Parkin *et al.* [46] and Currant [41] shows that horse and red deer post-crania occur in a ratio of about 5:1 [46]; if teeth are included, the ratio is closer to 6:1. These ratios are almost the same as the ratio of knapping tools made on horse bones (and teeth) to those made on red deer. Consequently, the horse and red deer bones discarded at the site could have provided most (if not all) of the bones and teeth that were used as knapping tools inside the cave. We can speculate as to whether the bones were collected and stockpiled for later use as knapping tools.

Aspects of form and function are exemplified by the use of the horse incisor and molar for pressure-flaking and the metapodials for hammer blows. Enamel is the strongest organic material available to Magdalenian knappers, and the use of teeth as pressure-flaking tools combines this property of toughness and resilience with the size and shape of the individual teeth, which were small enough to be held easily in the palm of the hand for fine manipulation for precise retouching. The form of horse and red deer metapodials, on the other hand, is ideally suited to their use as knapping hammers. They are robust bones with thick cortical walls and straight shafts, which can be held easily as 'hammers' for use in detaching flakes with light and more powerful blows.

Fig 15 provides a summary of the knapping tasks undertaken at Gough's Cave; it shows how the different types of organic knapping tools may have been utilized to remove blanks from cores and modify the blades and bladelets to make the range of flint tools that characterizes the Gough's Cave lithic assemblage [13].

## Knapping tools in Magdalenian contexts

We review the evidence from 15 Magdalenian published sites where organic knapping tools were recorded. The geographical distribution of the sites is shown on Fig 16, and a summary of the assemblages is given in Table 4. Our analysis aims to determine whether there are common patterns among Magdalenian assemblages in the type of organic retouchers and percussors used, the knapping technique adopted and the distribution of these tools within a site.

While detailed descriptions of Magdalenian knapping tools are not always available in the literature, some similarities and differences can be observed between the Gough's Cave assemblage and the material from the sites listed in Table 4. For instance, the bone elements selected by the Magdalenian toolmakers at Roc-de-Marcamps (France) were diverse, with retouchers made from long bone diaphysis fragments, fragments of metapodials, rib and mandibles [106]. Long bone fragments dominated the assemblage of bone knapping tools from Isturitz (France), as well as retouchers made on broken metacarpals [96]. At the open-air site of Pincevent (France), the two retouchers identified within the assemblage were both long bone mesial fragments [105]. With the exception of the complete teeth, metatarsal and proximal phalanx from Gough's Cave, we are not aware of any other records of complete bones used as knapping tools from a Magdalenian context. The diversity of materials used as knapping percussors

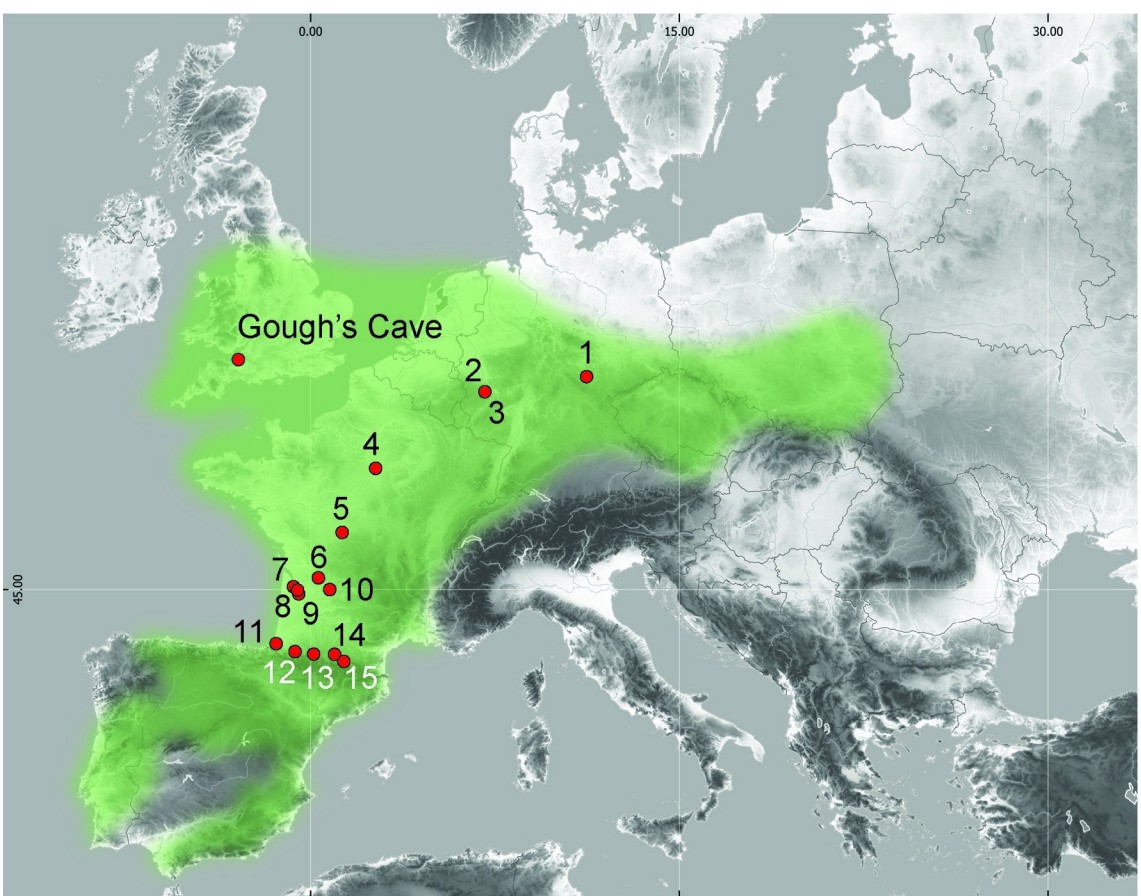

**Fig 16. Location of Gough's Cave and other Magdalenian sites with organic knapping tools in relation to the general distribution of Magdalenian sites (shown in green, after [90–93] and [94, Fig 8.3, p.439]). Key**: 1. Oelknitz; 2. Andernach; 3. Gönnersdorf; 4 Pincevent; 5. La Garenne; 6. Rochereil; 7. Roc-de-Marcamps; 8. Saint-Germain-de-la-Rivière; 9. Moulin-Neuf; 10. Laugerie-Haute; 11. Isturitz; 12. Laa 2 Cave; 13. Labastide; 14. Enlène; 15. La Vache. Map made with Natural Earth. Free vector and raster map data @ naturalearthdata.com.

during the Magdalenian is illustrated at sites such as Gönnersdorf [110, 111, 113, 114] and Oelknitz [112] in Germany, and La Vache, in France [100, 101], where bone or ivory as well as pebbles were used for knapping. Evidence from the French sites of Enlène [95] and Laugerie-Haute [1] indicates that antlers were also modified to make knapping hammers during the Magdalenian.

Bone assemblages from a selection of the more-recently excavated Magdalenian sites provide further insights into Magdalenian knapping tools and their prevalence in different site contexts. Geographically, the nearest informative sites are Gönnersdorf and Andernach-Martinsberg, two of the major Magdalenian settlement sites on opposite banks of the Rhine in the Neuwied Basin, Germany. Both sites are particularly well preserved, largely due to burial beneath volcanic deposits of the Laacher See eruption. Magdalenian occupation at these sites occurred simultaneously (between about 16,300 and 14,400 yrs cal BP) and prior to the warming of Greenland Interstadial GI 1e [115]. Street and Turner [116] analysed the substantial bone assemblage from Gönnersdorf and identified only 16 knapping tools in a bone assemblage dominated by horse bones (n = 8,656 identified specimens) with a lesser contribution from reindeer [110, 116, 117]. Mapping of the finds showed that the retouchers are spatially limited to a small area of the site interpreted as a dwelling structure (structural unit K II). The

**Table 4. Organic knapping tools reported for various Magdalenian bone assemblages.**

| No. on map (Fig 16) | Site | Bone | Tooth/Ivory | Antler | Animal species | Anatomical elements | Fragment | Complete | References |
|---|---|---|---|---|---|---|---|---|---|
| | Gough's Cave, UK | 7 | 2 | | *Equus ferus* (n = 8), *Cervus elaphus* (n = 1) | Third molar, incisor, metapodials, phalanx Metatarsal | 5 | 4 | This paper |
| 14 | Enlène, France | | | 1 | *Rangifer tarandus* | Antler (basal part) | | | [95] |
| 11 | Isturitz, France | 131 | | | *Rangifer tarandus, Equus ferus, Cervus elaphus,* bovids | Mostly tibia, humerus, femur, metacarpal | | | [96–98] |
| 5 | La Garenne, France | | | | | | | | [99] |
| 15 | La Vache, France | 55 | | | | Long bone diaphysis | | | [100, 101] |
| 12 | Laa 2 Cave, France | 2 | | | Middle size ungulates | Radius diaphysis | 2 | 0 | [102] |
| 13 | Labastide, France | 10 | | | | | | | [103] |
| 10 | Laugerie-Haute, France | | | 1 | *Cervus elaphus* | Shed antler with basal tines removed together with the beam and crown above the bez tine | | | [1] |
| 9 | Moulin-Neuf, France | 1 | | | | Long bone diaphysis | 1 | | [104] |
| 5 | Pincevent, France | 2 | | | | Left tibia mesial fragment; left radioulna mesial fragment | 2 | | [105] |
| 7 | Roc-de-Marcamps, France | 14 | | | *Rangifer tarandus, Saiga tatarica, Equus ferus,* bovid | Long bone diaphysis, metatarsal diaphysis, metacarpal diaphysis, mandible, rib | 14 | 0 | [106] |
| 6 | Rochereil, France | 2 | | | | Rib with engraving, undescribed bone with engraving | 1 | | [107, 108] |
| 8 | Saint-Germain-la-Rivière, France | 6 | | | Large ungulates (n = 3), *Saiga tatarica* (n = 2), bovid (n = 1) | Long bone diaphysis | 6 | | [109] |
| 2 | Andernach, Germany | 10 | | | | | | | [110, 111] |
| 3 | Gönnersdorf, Germany | 11 | | | | | | | [110, 111] |
| 1 | Oelknitz, Germany | | 1 | | | Ivory fragment | 1 | | [112] |

Details from publications are often incomplete. Where known, the type of raw material (bone, tooth/ivory, antler) and the type of blank (fragment or complete) are tabulated.

Gönnersdorf knapping tools are interpreted as *ad hoc* implements and all but one (a rib) are made on fragments of horse long-bones. As with the Gough's Cave assemblage, the knappers selected horse metapodials for flint working, but in contrast to the Gough's Cave examples, those from Gönnersdorf are on metapodial shafts fragments (n = 4) or split shafts retaining part of the proximal articulation (n = 5). The knapping damage on these pieces is invariably located near the broken end of the fragments, and six pieces have damage located at both ends. The rib is interpreted as a possible pressure-flaking tool, whereas the other tools are interpreted as percussors [116]. Gönnersdorf provides a good example of spatial organisation at a Magdalenian campsite with a clear pattern of specialised activity areas linking the knapping tools to the place where stone tools were worked.

The recently excavated bone assemblage from Laa 2 Cave in the foothills of the French Pyrenees provides another well-recorded Magdalenian faunal assemblage [102]. The results of the test-pitting investigations show that the cave was occupied by Magdalenian hunters-gatherers between about 20,000 and 15,000 cal BP. Only two retouchers (the one illustrated is a radius shaft fragment; see Fig 21 in [102]) were identified in this large faunal assemblage

dominated by ungulate remains (n = 932: in order of decreasing abundance these are horse, reindeer, ibex, chamois, red deer, bison and roe deer).

Similarly, only two retouchers were identified within the large faunal assemblage from the open-air site of Pincevent (Seine-et-Marne, France), where the remains of a 13,000 years-old Magdalenian camp occupying a vast area of nearly 5,000 m$^2$ were discovered in 1964, on the banks of the Seine river [105, 118, 119]. Detailed analyses of the flint and fauna refittings suggest a complex social organization, with several habitations and workshops [118]. The activities carried out by the Magdalenians from Pincevent mainly concentrated around the hunting and processing of at least 71 reindeers, used for meat consumption as well as for raw materials, as illustrated by artefacts such as a reindeer antler point fragment with embedded flint barbs, likely used as a spear [120, 121]. Antlers appear to have been selected primarily to be worked and used as tools, with less than twenty bone tools (e.g. needle fragments, lissoirs, etc.) identified in the assemblage [122]. The two retouchers reported from Pincevent are mesial long bone fragments, one bearing crushing marks on its postero-lateral edge, and the other displaying fine subvertical scratches on its antero-lateral edge as well as underneath a gnawed surface [105].

A particularly intriguing set of retouchers was found in a Magdalenian level at Rochereil (France). These include a rib decorated with a schematic animal figure illustrated by Rémy [107] (Fig 112.3, p.308, in [107]). Details are scant, but Rémy (ibid.) mentions the presence in the same deposits of bone tools, such as needles, lissoirs and retouchers. Another study focussing on the portable art from the site mentions two further bone retouchers bearing engravings [108].

Three further quantified assemblages provide further support for the suggestion that organic knapping tools may be a rare artefact type in Magdalenian bone-tool assemblages. The first is the Final Magdalenian horizon at Solutré (Burgundy, France), dated to 15,080 +/- 130 BP [123]. This site is characterized by a vast quantity of bone debris deposited at kill-sites where horses (supplemented by reindeer and bison) were hunted and processed. The large mammal assemblage from one small area of this extensive site was studied in detail by Turner [123, 124]. She undertook a thorough taphonomic study of this cultural level (in sector P16), examining more than 4,000 large mammal bones for traces of human modification. She confirmed a surprisingly low prevalence of traces of butchery, suggesting that, although large numbers of horses were killed at any one time, their carcasses were not fully exploited. Another unusual feature of sector P16 assemblage is the presence of several different types of bone artefacts; these include a needle core, a perforated *bâton*, a fragmentary double bevelled-based point and some waste from antler working. No organic knapping tools were identified in this assemblage. This perhaps points to the somewhat specialized nature of the Magdalenian activities in Sector P16. In a wholly domestic setting or a site where stone tools were manufactured or repaired, the frequency of bone tools would probably be higher.

The second site is the cave of Le Placard (Charente, France). This exceptionally rich Magdalenian site has yielded substantial quantities of carefully-shaped osseous artefacts. A recent study of two unpublished collections by Langley and Delage [125] suggests that museum collections may be biased in terms of the type of artefacts curated as well as selective publication of more easily-recognizable tool types.

Finally, the cave of Grotte des Eyzies (southwest France) provides a further example of a Magdalenian site at which knapping tools appear to be absent in a large collection which has been the subject of a detailed taphonomic analysis. Excavated in the mid-nineteenth century, the cave contained rich occupation layers corresponding to Magdalenian occupation. The cave was seasonally-occupied and used as a camp principally for the processing of reindeer carcasses for storage by means of drying or smoking of filleted meat [126]. The site was excavated

by Lartet and Christy [127], who removed large blocks of breccia which were distributed to museums throughout France, Britain and elsewhere in the world. Olsen studied the faunal remains from mechanically breaking down the bone- and artefact-rich breccia blocks housed at the British Museum. As well as flint tools and hammerstones, the blocks yielded a diverse range of finds, including stone lamps, engraved stone, mobiliary art on bone, worked ivory, fine bone needles, antler spear and harpoon points [126]. The typology of the artefacts places the site in the Late Magdalenian. Olsen's study supports the interpretation of a reindeer-based economy (n = 833 identified bones), with large bovids (n = 12) and horses (n = 17) as the other principal large mammals exploited. Although many of the bones have cut marks and signs of marrow breakage, and the assemblage includes a high proportion of shaft fragments, no knapping tools were identified by Olsen.

The frequency of knapping tools in the Gough's Cave assemblage (relative to the number of bones) is difficult to estimate because the skeletal element representation in the NHM collection is heavily distorted due to the disposal of many of the bone fragments for curatorial reasons [41]. Nevertheless, metapodials and phalanges are well represented, and these elements are ones that were retained. In total, the Gough's Cave assemblage includes 12 distal metapodials (metacarpal III, metatarsal III), of which 41.7% (n = 5) have percussive damage from knapping. In comparison, only 5% (n = 1) of the horse proximal phalanges (n = 21) exhibit knapping damage. This difference may be due to the shorter 'life' of the metapodial percussors, which broke during use and had to be replaced, whereas the phalanx is complete and may have been used on several occasions.

In their study of the Upper Palaeolithic assemblages from the Swabian Jura (Germany), Toniatio *et al.* [128] noticed significant differences in the composition of the knapping tool assemblage from the Aurignacian horizons, which are characterized by a greater variety of bone elements (long bones, ribs, ivory, antler and carnivore teeth) than those from the Gravettian and Magdalenian levels, where very few organic retouchers were recovered. This apparent decline in the presence of organic retouching tools across the Upper Palaeolithic was hypothesized to relate to an increased use of stone hammers, likely due to changes in weaponry and ornamentation which required a shift in the raw materials selected [128, 129]. Whether this pattern holds more generally is unclear as the same types of knapping tool are found in the Magdalenian period—these includes bone fragments used as retouchers, complete bones used as hammers, antler soft hammers, teeth used as retouchers and stone hammers (Table 4). Searching for patterns relating to broader trends in knapping technology are unfortunately hampered by several factors, not solely relating to the relatively small number of Magdalenian sites where knapping tools have been recovered and quantified. At other sites, there is a problem of recognition, as *ad-hoc* tools are difficult to identify because they were both expediently used over short periods of time and the knapping marks may be subtle, but even antler hammers have been missed in previous analyses in some collections [1].

The examples used in our survey of Magdalenian sites show that there was spatial partitioning of activities with finds of knapping tools coming almost exclusively from areas of the site where the knapping was undertaken (e.g. Gönnersdorf). The absence of knapping tools at other sites may be related to the limited range of activities undertaken at specialist sites (e.g. kill-butchery locale of Solutré) or simply due to random factors such as sampling effects (Grotte des Eyzies).

## Conclusions

An intriguing aspect of Magdalenian knapping technology is the overall rarity of organic stone-working tools at Magdalenian cave and open-air sites. This is particularly conspicuous

for the type of *ad hoc* retouchers made on bone fragments that dominate earlier knapping-tool kits [2]. This rarity of organic knapping tools has been linked to an increased use of hard hammers and 'stone retouchers' [*sensu* 129, 130], however, other explanations, not necessarily mutually exclusive, should be considered.

The Magdalenian is characterised by its particularly rich and diverse variety of specialised organic tools, such as lissoirs, perforated batons, barbed points, harpoons, and needles, as well as exceptional mobile artworks, which have focussed archaeological interest. This focus has undoubtedly 'masked' awareness of 'unspecialized' tools in the Magdalenian technological repertoire; knapping tools are in this category and probably have been overlooked in many assemblages for this reason alone.

A significant contributing factor is undoubtedly the difficulty of identifying knapping damage on bones that have seen only a short period of use. This is exemplified by our discovery in the Gough's Cave collection of bones and teeth used as knapping tools that were overlooked in earlier comprehensive studies of the faunal collection [38, 46]. More recently, a flurry of publications has helped archaeologists and faunal specialists to recognize the diversity of 'retouchers' and the type of damage associated with their use. This has benefited from developments in imaging methods, most notably 3D optical microscopy and EDX to identify microscopic lithic inclusions embedded in the working area of knapping tools [e.g. 2, 63]. It is now apparent that organic knapping tools are more common in Lower, Middle and Early Upper Palaeolithic assemblages than formerly appreciated.

Regardless of the reason(s), it is worth noting, that, although the number of knapping tools found at Gough's Cave and other Magdalenian sites is relatively small, they contribute to our understanding of broader issues relating to the techniques and processes of knapping, and the overall development of technological innovations during the Magdalenian period. Integrating these 'unspecialized' knapping tools in the overall interpretation of Magdalenian knapping processes and site organization sheds new light on the activities undertaken at Gough's Cave. For example, the spatial distribution of the knapping tools at Gough's Cave, as well as at other Magdalenian sites, helps to locate areas dedicated to the production and maintenance of stone tools. It also extends knowledge about the variety of raw materials (pebble, bone, antlers, teeth, and ivory) and bone elements selected by Magdalenian hunters as useful objects that were involved in more complex processing activities, rather than discarded as waste. The Gough's Cave knapping tools also illustrate aspects of curation (the horse molar) and *ad hoc* (single-use?) tool-use, exemplified by the horse metapodials used as knapping hammers. Rather than simply dismissing such artefacts as expedient ('recycled') bone tools, the 'unspecialized' Magdalenian knapping tools merit in-depth analyses; after all, they are the starting point in the production of stone and other organic artefacts. These are clearly a key element in a technology that accompanied Magdalenian hunters as they adapted and expanded into new territories across northern and central Europe during a period of rapid and dramatic climate change at the start of the Lateglacial Interstadial.

## Acknowledgments

Analysis of the Gough's Cave material was made possible thanks to the generosity of the Longleat Estate and their long-term loan (NHM loan: PAL 2020–498 PV) of finds to the Natural History Museum (NHM). We are also indebted to the former and current curators and colleagues at the NHM, especially Andy Currant, Robert Kruszynski, Roula Pappa, Rachel Ives, Heather Bonney, Pip Brewer and Chris Stringer; their expertise, knowledge and insights were invaluable in making sense of the Gough's Cave collections at the NHM. Linzi Harvey compiled a comprehensive catalogue of the finds from Gough's Cave, and conservation and re-

boxing of the collection was overseen by Effie Verveniotou and colleagues in the NHM Conservation Centre. We are thankful to Monika Knul and Linzi Harvey for collating the Gough's Cave radiocarbon dates, and to Hugh Parfitt for his assistance with the drafting of Fig 14. Finally, we would like to thank the staff at the Image Suite at the NHM, in particular Alex Ball, Amy Scott-Murray and Innes Clatworthy, for their continuing efforts to make microscopy facilities at the museum accessible, even remotely during the Covid pandemic. We are indebted to the Calleva Foundation for supporting our work. J.G.W's work is supported by the National Science Foundation Graduate Research Fellowship. Finally, we thank our reviewers and the editor for helpful comments and suggestions which have improved the quality of our paper.

## Author Contributions

**Conceptualization:** Silvia M. Bello, Simon A. Parfitt.

**Data curation:** Silvia M. Bello, Lucile Crété, Julia Galway-Witham, Simon A. Parfitt.

**Formal analysis:** Silvia M. Bello, Simon A. Parfitt.

**Funding acquisition:** Silvia M. Bello.

**Investigation:** Silvia M. Bello, Simon A. Parfitt.

**Methodology:** Silvia M. Bello.

**Project administration:** Silvia M. Bello.

**Resources:** Simon A. Parfitt.

**Supervision:** Silvia M. Bello, Simon A. Parfitt.

**Validation:** Simon A. Parfitt.

**Visualization:** Silvia M. Bello, Simon A. Parfitt.

**Writing – original draft:** Silvia M. Bello, Simon A. Parfitt.

**Writing – review & editing:** Silvia M. Bello, Lucile Crété, Julia Galway-Witham, Simon A. Parfitt.

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
