## [Decision Letter · Decision Letter 0]

31 Aug 2021

PONE-D-21-20090

Knapping tools in Magdalenian contexts: New evidence from Gough’s Cave (Somerset, UK)

PLOS ONE

Dear Dr. Bello,

Thank you for submitting your manuscript to PLOS ONE. After careful consideration, we feel that it has merit but does not fully meet PLOS ONE’s publication criteria as it currently stands. Therefore, we invite you to submit a revised version of the manuscript that addresses the points raised during the review process.

We look forward to receiving your revised manuscript.

Kind regards,

Enza Elena Spinapolice, Ph.D

Academic Editor

PLOS ONE

Journal Requirements:

2. In your manuscript, please provide additional information regarding the specimens used in your study. Ensure that you have reported specimen numbers and complete repository information, including museum name and geographic location.

For more information on PLOS ONE's requirements for paleontology and archaeology research, see https://journals.plos.org/plosone/s/submission-guidelines#loc-paleontology-and-archaeology-research.

3. We note that Figure(s) 1, 12 and 16 in your submission contain [map/satellite] images which may be copyrighted. All PLOS content is published under the Creative Commons Attribution License (CC BY 4.0), which means that the manuscript, images, and Supporting Information files will be freely available online, and any third party is permitted to access, download, copy, distribute, and use these materials in any way, even commercially, with proper attribution. For these reasons, we cannot publish previously copyrighted maps or satellite images created using proprietary data, such as Google software (Google Maps, Street View, and Earth). For more information, see our copyright guidelines: http://journals.plos.org/plosone/s/licenses-and-copyright.

1. You may seek permission from the original copyright holder of Figure(s) 1, 12 and 16 to publish the content specifically under the CC BY 4.0 license.  

4. We note that Figure(s) 14 and 15 in your submission contain copyrighted images. All PLOS content is published under the Creative Commons Attribution License (CC BY 4.0), which means that the manuscript, images, and Supporting Information files will be freely available online, and any third party is permitted to access, download, copy, distribute, and use these materials in any way, even commercially, with proper attribution. For more information, see our copyright guidelines: http://journals.plos.org/plosone/s/licenses-and-copyright.

1. You may seek permission from the original copyright holder of Figure(s) 14 and 15 to publish the content specifically under the CC BY 4.0 license. 

Reviewers' comments:

Reviewer's Responses to Questions

**Comments to the Author**

1. Is the manuscript technically sound, and do the data support the conclusions?

Reviewer #1: Yes

Reviewer #2: Yes

2. Has the statistical analysis been performed appropriately and rigorously? 

Reviewer #1: N/A

Reviewer #2: N/A

3. Have the authors made all data underlying the findings in their manuscript fully available?

Reviewer #1: Yes

Reviewer #2: Yes

4. Is the manuscript presented in an intelligible fashion and written in standard English?

Reviewer #1: Yes

Reviewer #2: Yes

5. Review Comments to the Author

Reviewer #1: In their paper, Bello and collaborators provide a detailed analysis of osseous soft hammers, retouchers and pressure flakers found at Gough’s Cave, a well-studied Magdalenian site that has yielded numerous human remains. This study is particularly relevant both theoretically and methodologically. From a theoretical perspective, ‘expedient’ osseous tools are poorly documented in Upper Palaeolithic context owing to the focus on ‘formal’ tools since the early development of archaeology. Indeed, ‘formal’ tools were easily recognizable and many types played a central role in defining the chrono-cultural framework of the European Upper Palaeolithic. Moreover, ‘expedient’ bone tools are somewhat hard to recognize and many post-depositional processes could result in surface alterations that mimic anthropogenic modifications. Fortunately, studies published in the last three decades have greatly expanded our understanding of these processes and archaeologists are now equipped with a set of tools allowing them to securely make such distinction. From a methodological standpoint, Bello and collaborators provide a rare example of how different microscopic approaches can be combined to support their functional interpretation. Collectively, their results and interpretations provide a new understanding of the complexity inherent to lithic technology during the Magdalenian. I sincerely believe this paper is worthy of publication in PLOS ONE, and recommend only a few minor suggestions, revisions, additions, and clarifications that the authors may want to consider before moving forward. They are listed below.

Lines 89-90: Although it is true that ‘organic (soft) knapping tools appear to be rare in Magdalenian contexts’, this could be also due to the use of Buxus sp. hammers.

Lines 161-164: It would be pertinent, in addition to providing the mean value, to include the range of the 14C ages obtain when dating the remains from Gough’s Cave.

Lines 166-169: It would be interesting to provide a summary of the evidence for seasonality at Gough’s Cave.

Material and methods: As a general comment, I would appreciate the authors to provide more references on which they based their interpretations on 1) post-depositional alteration, 2) typological attribution, 3) technological identification of use wear traces, 4) breakage patterns.

Lines 185-214: This section reads more like an overview of “Result and Discussion” rather the “Material and methods”. A more standardized presentation of the Material and methods used in the study would improve the flow of the paper.

Line 200: The first edition (Russian) of Semenov’s work was published in 1957.

Lines 213-214: The authors state that ‘the terminology for the different types of knapping tool used in the literature is […] confused.’ Although I agree with them, I wish they would expand more on the issue, and, most importantly, provide clear definition for the terminology they use in their manuscript, as well as how this compares to others, e.g., Mallye et al. 2012; Mozota 2012, 2018, etc.

Table 2: Please provide the references where the 14C ages were first published.

Line 235: ‘A sample of these features […]’. Please provide an explanation as to how and why this sample of features was selected.

Lines 240-241: Please provide a brief summary of the rationale for combining multiple imaging techniques. What type of information do they each provide?

Radiocarbon dates: It is not clear to me why the authors didn’t choose to present this information in the “Archaeological context” section. It is only as of line 257 that I realized the 14C ages had been previously published. Perhaps it would be best to reconsider when this paragraph appears in the text.

Abbreviations: It is not clear why this paragraph doesn’t appear as a Note for Table 2.

Line 324-327: Perhaps it would be good to also mention more recent experimental research on pressure flaking such as Nami and Scheinsohn 1997; Armand and Delagnes 1998; d’Errico et al. 2012; Mozota 2015; Doyon et al. 2019.

Lines 333-336: It would be interesting to explain why the authors favor this interpretation as oppose to Jéquier’s according to which these striations may have resulted from preparing the margins of the lithic tools before retouching it (Jéquier et al. 2018)

Lines 347-355: This paragraph reads more like a summary of the interpretations. It seems oddly placed as it comes before presenting the results themselves.

Lines 378-380: Some references supporting this claim would be welcomed.

Line 427: It would be interesting to have a clear image showing an impact that has been sectioned by a fracture and clarify if the fracture is fresh or weathered.

Lines 493-495: Please provide references to these experimental studies.

Lines 502-503: Please provide references describing the breakage pattern resulting from each activity and explain how both could be confounded.

Lines 548 & 558: The interpretation is counter-intuitive. If most knapping damages are concentrated close to the distal end, one would assume the phalanx was held at the proximal end. Furthermore, from the images, despite their low resolution, it would appear the contact with the flint starts from the proximal end toward the distal end and not the other way around. Please clarify.

Line 576: ‘examind’ should be ‘examined’

Line 612: six inches is the equivalent of ~15 cm, not five.

Line 693: ‘flacs’ should be ‘flancs’

Lines 734-735: Armand and Delagnes (1998) as well as Doyon et al. (2019) make a distinction between active and passive pressure flaking. In their paper, Doyon et al. seem to suggest it would generate different traces which seems to be supported by Mozota (2015) and d’Errico et al. (2012). The use of ‘passive retouching’ could probably be nuanced a little bit here.

Line 738: Wood can also be used to retouch stone tools. Perhaps citing Martellotta et al. 2021 would be a good idea here.

Lines 747-748: Some references of knapping experiments would be appreciated here.

Lines 788-789: The use of organic knapping tools in an ad hoc AND curated fashion has been documented elsewhere, e.g., Costamagno et al. 2018; Doyon et al. 2018. It might be worth citing these examples as well.

Lines 799-800: This is more a question than a suggestion, but would you have any idea as to why bone marrow would have been overlooked from these skeletal elements?

Line 816: Here too, Doyon et al. 2018 could be cited.

Line 847-849: Jéquier et al. 2018 argue these traces could be resulting from the preparation of the margins of the lithic tools before retouching.

Line 850: Roasting a bone would have made it quite fragile and not very useful for knapping. In Spain, for instance, shepherds are often roasting bones to access the marrow. When exposing a complete bone to the flame, it cracks in less than two or three minutes, easing the access to marrow.

Lines 953-954: It is not clear what is implied by the term ‘typical’ here. It could mean either ‘type fossil’ or ‘commonly found’. Needles, lissoirs and retouchers and found in many techno-cultures and therefore and not ‘typical’ of the Magdalenian in the sense of ‘type fossil’. It would perhaps be best to find another word here to avoid ambiguity.

Line 1032: At the end of this section, I was surprised not to see any comparison with the Pincevent site. Pincevent has been extensively excavated, with a clear demonstration of spatial partitioning of activities (Julien and Karlin 2014) and rare occurrence of bone retouchers (Leroi-Gourhan and Brézillon 1972), which collectively seem to fit what is known for Gough’s Cave.

Line 1044-1047: This is also the case for the Aurignacian, and most of the European Upper Palaeolithic.

References section. I noticed quite a few issues with capitalization in the references. The authors may want to review the guide for authors to ensure these are resolved in an updated version of their manuscript.

List of references cited in this review:

Armand, Dominique, and Anne Delagnes. “Les Retouchoirs En Os d’Artenac (Couche 6c): Perspectives Archéozoologiques, Taphonomiques et Expérimentales.” In Économie Préhistorique L Les Comportements de Subsistance Au Paléolithique. XVIIIe Rencontre Internationale d’archéologie et d’histoire d’Antibes, edited by Jean-Philippe Brugal, Liliane Meignen, and Marylène Patou-Mathis, 205–14. Antibes: Éditions APDCA, 1998.

Costamagno, Sandrine, Laurence Bourguignon, Marie-Cécile Soulier, Liliane Meignen, Cédric Beauval, William Rendu, Célimene Mussini, Alan Mann, and Bruno Maureille. “Bone Retouchers and Site Function in the Quina Mousterian: The Case of Les Pradelles (Marillac-Le-Franc, France).” In The Origins of Bone Tool Technologies, edited by Jarod M. Hutson, Alejandro García-Moreno, Elisabeth S. Noack, Elaine Turner, Aritza Villaluenga, and Sabine Gaudzinski-Windheuser, 165–95. Mainz: Römisch Germanisches ZentralMuseum, 2018.

d’Errico, Francesco, Lucinda R. Backwell, and Lyn Wadley. “Identifying Regional Variability in Middle Stone Age Bone Technology: The Case of Sibudu Cave.” Journal of Archaeological Science 39, no. 7 (July 1, 2012): 2479–95. https://doi.org/10.1016/j.jas.2012.01.040.

Doyon, Luc, Hao Li, ZhanYang Li, Hua Wang, and QingPo Zhao. “Further Evidence of Organic Soft Hammer Percussion and Pressure Retouch from Lingjing (Xuchang, Henan, China).” Lithic Technology 44, no. 2 (March 21, 2019): 100–117. https://doi.org/10.1080/01977261.2019.1589926.

Doyon, Luc, Zhanyang Li, Hao Li, and Francesco d’Errico. “Discovery of circa 115,000-Year-Old Bone Retouchers at Lingjing, Henan, China.” PLOS ONE 13, no. 3 (March 12, 2018): e0194318. https://doi.org/10.1371/journal.pone.0194318.

Jéquier, Camille, Alessandra Livraghi, Matteo Romandini, and Marco Peresani. “Same but Different: 20,000 Years of Bone Retouchers from Northern Italy. A Diachronologic Approach from Neanderthals to Anatomically Modern Humans.” In The Origins of Bone Tool Technologies, edited by Jarod M. Hutson, Alejandro García-Moreno, Elisabeth S. Noack, Elaine Turner, Aritza Villaluenga, and Sabine Gaudzinski-Windheuser, 269–85. Mainz: Römisch Germanisches ZentralMuseum, 2018.

Julien, Michèle, and Claudine Karlin, eds. Un Automne à Pincevent. Le Campement Magdalénien Du Niveau IV20. Mémoire de La Société Préhistorique Française 57. Paris, France: Société préhistorique française, 2014.

Leroi-Gourhan, André, and Michel Brézillon. “Fouilles de Pincevent. Essai d’analyse ethnographique d’un habitat magdalénien (la section 36).” Gallia Préhistoire 7, no. 1 (1972). https://www.persee.fr/doc/galip_0072-0100_1972_sup_7_1.

Mallye, Jean-Baptiste, Céline Thiébaut, Vincent Mourre, Sandrine Costamagno, Émilie Claud, and Patrick Weisbecker. “The Mousterian Bone Retouchers of Noisetier Cave: Experimentation and Identification of Marks.” Journal of Archaeological Science 39, no. 4 (April 2012): 1131–42. https://doi.org/10.1016/j.jas.2011.12.018.

Martellotta, Eva Francesca, Jayne Wilkins, Adam Brumm, and Michelle C. Langley. “New Data from Old Collections: Retouch-Induced Marks on Australian Hardwood Boomerangs.” Journal of Archaeological Science: Reports 37 (June 1, 2021): 102967. https://doi.org/10.1016/j.jasrep.2021.102967.

Mozota Holgueras, Millán. “El Hueso Como Material Prima: El Utillage Óseo Del Final Del Musteriense En El Sector Central Del Norte de La Península Ibérica.” Ph.D., Universidad de Cantabria, 2012.

———. “Un Análisis Tecno-Funcional de Los Retocadores Óseos Musterienses Del Norte de La Península Ibérica, y Su Aplicación al Estudio de Los Grupos Neandertales.” Munibe Antropologia - Arkeologia 66 (2015): 5–21. https://doi.org/10.21630/maa.2015.66.01.

———. “Experimental Programmes with Retouchers: Where Do We Stand and Where Do We Go Now?” In The Origins of Bone Tool Technologies, edited by Jarod M. Hutson, Alejandro García-Moreno, Elisabeth S. Noack, Elaine Turner, Aritza Villaluenga, and Sabine Gaudzinski-Windheuser, 15–32. Mainz: Römisch Germanisches ZentralMuseum, 2018.

Nami, Hugo G., and Vivian G. Scheinsohn. “Use-Wear Patterns on Bone Experimental Flakers: A Preliminary Report.” In Proceedings of the 1993 Bone Modification Conference, Hot Springs, South Dakota, edited by L. Hannus, L. Rossum, and P. Winham, 256–64. Sioux Falls, SD: Archeology Laboratory, Augustana College, 1997.

Tartar, Élise. “Réflexion Autour de La Fonction Des Retouchoirs En Os de l’Aurignacien Ancien.” Bulletin de La Société Préhistorique Française 109, no. 1 (2012): 69–83.

———. “The Recognition of a New Type of Bone Tools in Early Aurignacian Assemblages: Implications for Understanding the Appearance of Osseous Technology in Europe.” Journal of Archaeological Science 39, no. 7 (July 2012): 2348–60. https://doi.org/10.1016/j.jas.2012.02.003.

Reviewer #2: The paper submitted is without major issues. It is of real interest. Here are some minor comments on the manuscript.

l64-66: I do not clearly understand if this information is usefull here? I suggest to delete this sentence.

l86-87: Maybe provide some references on studies where scholars precisely identify the technic of percussion and nature of soft hammer. Furthermore, you provide no indication on the Magdalenian lithic industry of Gough’s Cave: is there no references supporting use of soft organic percussion for at least par of the material?

l146-147: I do not understand if this assumption is supported only by anthropological data or if other data support it. If so, maybe provide some references to other arguments.

l216-219: It could be useful to provide the numbers of pieces initially selected after naked eye examination and the proportion of confirmed specimen after binocular and microscopical examination. .

l374-376: I did not understood if there are reasons to do not clean pieces to remove this sediment (why not using cleaning bath for example?). Maybe explain that.

l493-496: Please provide references about these experimental studies

6. PLOS authors have the option to publish the peer review history of their article (what does this mean?). If published, this will include your full peer review and any attached files.

Reviewer #1: No

Reviewer #2: No

---

## [Author Response · Author response to Decision Letter 0]

7 Oct 2021

Dr Silvia Bello

Dept. Earth Science

The Natural History Museum

Cromwell Road

London SW7 5BD, UK

s.bello@nhm.ac.uk

PLOS-ONE Editorial Board 

7th October 2021

PLOS-ONE Editorial Board,

We would like to thank the Editorial Board and the two anonymous reviewers for their positive comments and valuable suggestions. We have revised the manuscript accordingly and corrected it to fully meet PLOS ONE’s publication criteria.

We present the full list of how we address the reviews below, with our comments highlighted in red.

We hope you can consider the paper ready for publication. Please do not hesitate to contact me in case you need further changes, corrections or files.

All best wishes

Silvia Bello

 

Journal Requirements:

Please review your reference list to ensure that it is complete and correct. If you have cited papers that have been retracted, please include the rationale for doing so in the manuscript text, or remove these references and replace them with relevant current references. Any changes to the reference list should be me¬ntioned in the rebuttal letter that accompanies your revised manuscript. If you need to cite a retracted article, indicate the article’s retracted status in the References list and also include a citation and full reference for the retraction notice.

AUTHORS COMMENTS: Style requirements are met in the manuscript.

2. In your manuscript, please provide additional information regarding the specimens used in your study. Ensure that you have reported specimen numbers and complete repository information, including museum name and geographic location.

For more information on PLOS ONE's requirements for paleontology and archaeology research, see https://journals.plos.org/plosone/s/submission-guidelines#loc-paleontology-and-archaeology-research.

AUTHORS COMMENTS: This has been clearly stated in the Section ‘Abbreviations’

3. We note that Figure(s) 1, 12 and 16 in your submission contain [map/satellite] images which may be copyrighted. All PLOS content is published under the Creative Commons Attribution License (CC BY 4.0), which means that the manuscript, images, and Supporting Information files will be freely available online, and any third party is permitted to access, download, copy, distribute, and use these materials in any way, even commercially, with proper attribution. For these reasons, we cannot publish previously copyrighted maps or satellite images created using proprietary data, such as Google software (Google Maps, Street View, and Earth). For more information, see our copyright guidelines: http://journals.plos.org/plosone/s/licenses-and-copyright.

1. You may seek permission from the original copyright holder of Figure(s) 1, 12 and 16 to publish the content specifically under the CC BY 4.0 license. 

AUTHORS COMMENTS: 

Figure 1: A. Base map drawn by one of the authors by tracing over a standard Europe base map (Source: https://en.wikipedia.org/wiki/Europe#/media/File:Europe_polar_stereographic_Caucasus_Urals_boundary.svg by Ssolbergj; CC BY-SA 3.0). B. Plan re-drawn by the authors based on based on Donovan 1955; Jacobi, 2004, Stringer 2000

Figure 12: Plans re-drawn by the authors based on Donovan 1955; Jacobi, 2004, Stringer 2000.

Figure 16: Figure caption updated with source information: ‘Made with Natural Earth. Free vector and raster map data available at naturalearthdata.com’.

4. We note that Figure(s) 14 and 15 in your submission contain copyrighted images. All PLOS content is published under the Creative Commons Attribution License (CC BY 4.0), which means that the manuscript, images, and Supporting Information files will be freely available online, and any third party is permitted to access, download, copy, distribute, and use these materials in any way, even commercially, with proper attribution. For more information, see our copyright guidelines: http://journals.plos.org/plosone/s/licenses-and-copyright.

AUTHORS COMMENTS: Figures 14 and 15 were both drawn and created by one the authors and have not been previously published.

AUTHORS COMMENTS: All data are presented in the main text.

Reviewer’s Comments to the Author

Reviewer #1: In their paper, Bello and collaborators provide a detailed analysis of osseous soft hammers, retouchers and pressure flakers found at Gough’s Cave, a well-studied Magdalenian site that has yielded numerous human remains. This study is particularly relevant both theoretically and methodologically. From a theoretical perspective, ‘expedient’ osseous tools are poorly documented in Upper Palaeolithic context owing to the focus on ‘formal’ tools since the early development of archaeology. Indeed, ‘formal’ tools were easily recognizable and many types played a central role in defining the chrono-cultural framework of the European Upper Palaeolithic. Moreover, ‘expedient’ bone tools are somewhat hard to recognize and many post-depositional processes could result in surface alterations that mimic anthropogenic modifications. Fortunately, studies published in the last three decades have greatly expanded our understanding of these processes and archaeologists are now equipped with a set of tools allowing them to securely make such distinction. From a methodological standpoint, Bello and collaborators provide a rare example of how different microscopic approaches can be combined to support their functional interpretation. Collectively, their results and interpretations provide a new understanding of the complexity inherent to lithic technology during the Magdalenian. I sincerely believe this paper is worthy of publication in PLOS ONE, and recommend only a few minor suggestions, revisions, additions, and clarifications that the authors may want to consider before moving forward. They are listed below.

Lines 89-90: Although it is true that ‘organic (soft) knapping tools appear to be rare in Magdalenian contexts’, this could be also due to the use of Buxus sp. hammers.

AUTHORS COMMENTS: This speculation is interesting – unfortunately it cannot be addressed because there are no Palaeolithic wooden hammers. We do, however, mention wood hammers as part of experimental knapping kits. Wood hammers are not included in the list (lines 737-9) of Palaeolithic knapping tools as there are no published examples from a Palaeolithic context.

Lines 161-164: It would be pertinent, in addition to providing the mean value, to include the range of the 14C ages obtain when dating the remains from Gough’s Cave.

AUTHORS COMMENTS: We do provide the range of 14C ages and provide a link to the most up-to-date paper with the full set of dates. ‘The dates display a remarkably tightly clustered group with a mean value of 12,600 BP (~14,950 – 14,750 cal BP) indicating that the Magdalenian occupation lasted for as little as two or three human generations [39]’

Lines 166-169: It would be interesting to provide a summary of the evidence for seasonality at Gough’s Cave.

AUTHORS COMMENTS: We have added a summary of the evidence for seasonality with the supporting reference (13).

Material and methods: As a general comment, I would appreciate the authors to provide more references on which they based their interpretations on 1) post-depositional alteration, 2) typological attribution, 3) technological identification of use wear traces, 4) breakage patterns.

AUTHORS COMMENTS: This has now been added in the text.

Lines 185-214: This section reads more like an overview of “Result and Discussion” rather the “Material and methods”. A more standardized presentation of the Material and methods used in the study would improve the flow of the paper.

AUTHORS COMMENTS: We have added a description and references for the methods used to interpret modifications of the faunal remains.

The ‘Material and methods’ section has now been re-written and presented in a more ‘conventional’ way. The text presented in the section has been moved to Results.

Line 200: The first edition (Russian) of Semenov’s work was published in 1957.

AUTHORS COMMENTS: Corrected to ‘1950s’

Lines 213-214: The authors state that ‘the terminology for the different types of knapping tool used in the literature is […] confused.’ Although I agree with them, I wish they would expand more on the issue, and, most importantly, provide clear definition for the terminology they use in their manuscript, as well as how this compares to others, e.g., Mallye et al. 2012; Mozota 2012, 2018, etc.

AUTHORS COMMENTS: This sentence has now been removed and a clearer terminology has been added to the methodology section.

Table 2: Please provide the references where the 14C ages were first published.

AUTHORS COMMENTS: We have added references to the primary references in the caption and table heading. ‘Other samples’ has been replaced with more specific information on sampling for aDNA.

Line 235: ‘A sample of these features […]’. Please provide an explanation as to how and why this sample of features was selected.

AUTHORS COMMENTS: The sentence has been modified to explain that ‘details of these features’ were recorded using the SEM.

Lines 240-241: Please provide a brief summary of the rationale for combining multiple imaging techniques. What type of information do they each provide?

AUTHORS COMMENTS: The sentence has been modified to accommodate this suggestion.

Radiocarbon dates: It is not clear to me why the authors didn’t choose to present this information in the “Archaeological context” section. It is only as of line 257 that I realized the 14C ages had been previously published. Perhaps it would be best to reconsider when this paragraph appears in the text.

AUTHORS COMMENTS: We have changed the heading to ‘Radiocarbon dating of the knapping tools’. This is part of the results as this is the first time radiocarbon dates have been linked to knapping tools at Gough’s Cave. This is an important result as the dates provide a direct link between these tools to the Magdalenian occupation. This section has now been added under the heading ‘Results’.

Abbreviations: It is not clear why this paragraph doesn’t appear as a Note for Table 2.

AUTHORS COMMENTS: This was added according to PLOS ONE's requirements for palaeontology and archaeology research.

Line 324-327: Perhaps it would be good to also mention more recent experimental research on pressure flaking such as Nami and Scheinsohn 1997; Armand and Delagnes 1998; d’Errico et al. 2012; Mozota 2015; Doyon et al. 2019.

AUTHORS COMMENTS: We would like to thank the reviewer for suggesting these references which we had overlooked. These have now been read and added to the manuscript.

Lines 333-336: It would be interesting to explain why the authors favor this interpretation as oppose to Jéquier’s according to which these striations may have resulted from preparing the margins of the lithic tools before retouching it (Jéquier et al. 2018)

AUTHORS COMMENTS: This has now been added for both the teeth retouchers.

Lines 347-355: This paragraph reads more like a summary of the interpretations. It seems oddly placed as it comes before presenting the results themselves.

AUTHORS COMMENTS: Most of this text has now been deleted.

Lines 378-380: Some references supporting this claim would be welcomed.

AUTHORS COMMENTS: The appropriate reference is:

Fernández-Jalvo, Y., and Andrews, P., 2016. Atlas of Taphonomic Identifications 1001+ Images of Fossil and Recent Mammal Bone Modification. Vertebrate Paleobiology and Paleoanthropology Series. Springer: Dordrecht, Heidelberg, New York, London.

Line 427: It would be interesting to have a clear image showing an impact that has been sectioned by a fracture and clarify if the fracture is fresh or weathered.

AUTHORS COMMENTS: Figure 6 has been modified and a clear photo of the impact has been added (figure 6 C),

Lines 493-495: Please provide references to these experimental studies.

AUTHORS COMMENTS: A reference has been provided.

Lines 502-503: Please provide references describing the breakage pattern resulting from each activity and explain how both could be confounded.

AUTHORS COMMENTS: The appropriate reference is:

Fernández-Jalvo, Y., and Andrews, P., 2016. Atlas of Taphonomic Identifications 1001+ Images of Fossil and Recent Mammal Bone Modification. Vertebrate Paleobiology and Paleoanthropology Series. Springer: Dordrecht, Heidelberg, New York, London.

Examples of broken metapodials from Gough’s Cave are described in this reference, together with the patterns of breakage inflicted by carnivores of different sizes. 

Lines 548 & 558: The interpretation is counter-intuitive. If most knapping damages are concentrated close to the distal end, one would assume the phalanx was held at the proximal end. Furthermore, from the images, despite their low resolution, it would appear the contact with the flint starts from the proximal end toward the distal end and not the other way around. Please clarify.

AUTHORS COMMENTS: We have corrected this mistake (line 558) to show that the phalanx was held at the proximal end, as shown in the reconstruction Fig 14B.

Line 576: ‘examind’ should be ‘examined’

AUTHORS COMMENTS: Corrected

Line 612: six inches is the equivalent of ~15 cm, not five.

AUTHORS COMMENTS: Corrected in text and Fig. 13 caption

Line 693: ‘flacs’ should be ‘flancs’

AUTHORS COMMENTS: Corrected

Lines 734-735: Armand and Delagnes (1998) as well as Doyon et al. (2019) make a distinction between active and passive pressure flaking. In their paper, Doyon et al. seem to suggest it would generate different traces which seems to be supported by Mozota (2015) and d’Errico et al. (2012). The use of ‘passive retouching’ could probably be nuanced a little bit here.

T AUTHORS COMMENTS: he text has been modified and the references added.

Line 738: Wood can also be used to retouch stone tools. Perhaps citing Martellotta et al. 2021 would be a good idea here.

AUTHORS COMMENTS: The Martellotta et al. 2021 reference has been added to line 724

Lines 747-748: Some references of knapping experiments would be appreciated here.

AUTHORS COMMENTS: References have been added

Lines 788-789: The use of organic knapping tools in an ad hoc AND curated fashion has been documented elsewhere, e.g., Costamagno et al. 2018; Doyon et al. 2018. It might be worth citing these examples as well.

AUTHORS COMMENTS: We checked Costamagno et l. 2018 paper, but we couldn’t find explicit references of curated tools. We noticed that Doyon et al., 2018 also mentioned the presence of ad hoc and curated tool at the Lingiing site. However, by writing that ‘..A similar spectrum from simple ad hoc knapping tools to curated items is known at least as far back as the early Middle Pleistocene Acheulean..’ we believe we cover other more recent sites.

Lines 799-800: This is more a question than a suggestion, but would you have any idea as to why bone marrow would have been overlooked from these skeletal elements?

AUTHORS COMMENTS: No, we can’t say.

Line 816: Here too, Doyon et al. 2018 could be cited.

AUTHORS COMMENTS: Interesting question, but this is something we would not be appropriate to address in this paper.

Line 847-849: Jéquier et al. 2018 argue these traces could be resulting from the preparation of the margins of the lithic tools before retouching.

AUTHORS COMMENTS: We address this point earlier in the text.

Line 850: Roasting a bone would have made it quite fragile and not very useful for knapping. In Spain, for instance, shepherds are often roasting bones to access the marrow. When exposing a complete bone to the flame, it cracks in less than two or three minutes, easing the access to marrow.

AUTHORS COMMENTS: The sentence has been modified to accommodate this suggestion.

Lines 953-954: It is not clear what is implied by the term ‘typical’ here. It could mean either ‘type fossil’ or ‘commonly found’. Needles, lissoirs and retouchers and found in many techno-cultures and therefore and not ‘typical’ of the Magdalenian in the sense of ‘type fossil’. It would perhaps be best to find another word here to avoid ambiguity.

AUTHORS COMMENTS: This has been resolved by deleting ‘typical of this techno-culture’.

Line 1032: At the end of this section, I was surprised not to see any comparison with the Pincevent site. Pincevent has been extensively excavated, with a clear demonstration of spatial partitioning of activities (Julien and Karlin 2014) and rare occurrence of bone retouchers (Leroi-Gourhan and Brézillon 1972), which collectively seem to fit what is known for Gough’s Cave.

AUTHORS COMMENTS: A discussion of the Pincevent site and material has been added in the discussion. 

Line 1044-1047: This is also the case for the Aurignacian, and most of the European Upper Palaeolithic.

AUTHORS COMMENTS: We have modified the sentence, but note that the range, diversity and richness of organic artefacts is a feature of the Magdalenian – other Upper Palaeolithic industries have similar types of artefacts, but the Magdalenian takes this to a new level.

Reviewer #2: The paper submitted is without major issues. It is of real interest. Here are some minor comments on the manuscript.

l64-66: I do not clearly understand if this information is usefull here? I suggest to delete this sentence.

AUTHORS COMMENTS: This sentence links the progression of stone working from the Middle to the Upper Palaeolithic – we modified the sentence and removed the following paragraph break to make this clear. 

l86-87: Maybe provide some references on studies where scholars precisely identify the technic of percussion and nature of soft hammer. Furthermore, you provide no indication on the Magdalenian lithic industry of Gough’s Cave: is there no references supporting use of soft organic percussion for at least part of the material?

AUTHORS COMMENTS: The technique of percussion is detailed later in the text, both in the methodology and discussion.

Reference to Magdalenian lithics is Jacobi reference [13]

l146-147: I do not understand if this assumption is supported only by anthropological data or if other data support it. If so, maybe provide some references to other arguments.

AUTHORS COMMENTS: This statement is based on archaeological data outlined in reference 13. We add this reference and a comment on seasonal use of the cave requested by Reviewer 1.

l216-219: It could be useful to provide the numbers of pieces initially selected after naked eye examination and the proportion of confirmed specimen after binocular and microscopical examination. .

AUTHORS COMMENTS: The text now includes a breakdown of the number of pieces examined. This is used to calculate the proportion of the different types of bone used as knapping tools. 

l374-376: I did not understood if there are reasons to do not clean pieces to remove this sediment (why not using cleaning bath for example?). Maybe explain that.

AUTHORS COMMENTS: We replace ‘adhering’ with ‘concreted’ and we explain why it has not been possible to clean the pieces more thoroughly. 

l493-496: Please provide references about these experimental studies

AUTHORS COMMENTS: A reference has been added.

---

## [Decision Letter · Decision Letter 1]

23 Nov 2021

Knapping tools in Magdalenian contexts: New evidence from Gough’s Cave (Somerset, UK)

PONE-D-21-20090R1

Dear Dr. Bello,

We’re pleased to inform you that your manuscript has been judged scientifically suitable for publication and will be formally accepted for publication once it meets all outstanding technical requirements.

Kind regards,

Enza Elena Spinapolice, Ph.D

Academic Editor

PLOS ONE

Reviewers' comments:

Reviewer's Responses to Questions

**Comments to the Author**

1. If the authors have adequately addressed your comments raised in a previous round of review and you feel that this manuscript is now acceptable for publication, you may indicate that here to bypass the “Comments to the Author” section, enter your conflict of interest statement in the “Confidential to Editor” section, and submit your "Accept" recommendation.

Reviewer #1: All comments have been addressed

Reviewer #2: All comments have been addressed

2. Is the manuscript technically sound, and do the data support the conclusions?

Reviewer #1: Yes

Reviewer #2: Yes

3. Has the statistical analysis been performed appropriately and rigorously? 

Reviewer #1: N/A

Reviewer #2: Yes

4. Have the authors made all data underlying the findings in their manuscript fully available?

Reviewer #1: Yes

Reviewer #2: Yes

5. Is the manuscript presented in an intelligible fashion and written in standard English?

Reviewer #1: Yes

Reviewer #2: Yes

6. Review Comments to the Author

Reviewer #1: The authors provide satisfactory responses to all the issues and questions raised by the reviewer. The changes introduced to this new version of the manuscript eliminate ambiguities and greatly improve the paper. A final proofread would allow the authors to remove some rare small mistakes in the text (e.g., line 924 with ref 94 cited twice; line 1039 with a space missing after the point, etc.) and make sure title capitalization is standardized throughout the bibliography. Aside from these minor adjustments, I strongly recommend the editor to accept the manuscript for publication.

Reviewer #2: (No Response)

7. PLOS authors have the option to publish the peer review history of their article (what does this mean?). If published, this will include your full peer review and any attached files.

Reviewer #1: No

Reviewer #2: No

---

## [Editor Report · Acceptance letter]

14 Dec 2021

PONE-D-21-20090R1 

Knapping tools in Magdalenian contexts:
New evidence from Gough’s Cave (Somerset, UK) 

Dear Dr. Bello:

I'm pleased to inform you that your manuscript has been deemed suitable for publication in PLOS ONE. Congratulations! Your manuscript is now with our production department. 

Kind regards, 

on behalf of

Dr. Enza Elena Spinapolice 

Academic Editor

PLOS ONE